# INSTRUCTSCENE: INSTRUCTION-DRIVEN 3D INDOOR SCENE SYNTHESIS WITH SEMANTIC GRAPH PRIOR

**Chenguo Lin, Yadong Mu**[*]
Peking University
`chenguolin@stu.pku.edu.cn, myd@pku.edu.cn`

## ABSTRACT

Comprehending natural language instructions is a charming property for 3D indoor scene synthesis systems. Existing methods directly model object joint distributions and express object relations implicitly within a scene, thereby hindering the controllability of generation. We introduce INSTRUCTSCENE, a novel generative framework that integrates a semantic graph prior and a layout decoder to improve controllability and fidelity for 3D scene synthesis. The proposed semantic graph prior jointly learns scene appearances and layout distributions, exhibiting versatility across various downstream tasks in a zero-shot manner. To facilitate the benchmarking for text-driven 3D scene synthesis, we curate a high-quality dataset of scene-instruction pairs with large language and multimodal models. Extensive experimental results reveal that the proposed method surpasses existing state-of-the-art approaches by a large margin. Thorough ablation studies confirm the efficacy of crucial design components. Project page: https://chenguolin.github.io/projects/InstructScene.

## 1 INTRODUCTION

Automatically synthesizing controllable and realistic 3D indoor scenes has been a persistent challenge for computer vision and graphics (Merrell et al., 2011; Fisher et al., 2015; Qi et al., 2018; Wang et al., 2018; Ritchie et al., 2019; Zhang et al., 2020; Yang et al., 2021b;a; Höllein et al., 2023; Song et al., 2023; Cohen-Bar et al., 2023; Lin et al., 2023; Feng et al., 2023; Patil et al., 2023). An ideal indoor scene synthesis system should fulfill at least three objectives: (1) comprehending instructions in natural languages, thus providing an intuitive and user-friendly interface; (2) designing object compositions that exhibit aesthetic appeal and thematic harmony; (3) placing objects in appropriate positions and orientations adhering to their functions and regular arrangements.

Natural instructions for interior design often rely on abstract object relationships, posing significant challenges for recent advancements in 3D scene synthesis (Wang et al., 2021; Paschalidou et al., 2021; Liu et al., 2023a; Tang et al., 2023) due to the implicit modeling of relationships through individual object attributes. Other studies (Luo et al., 2020; Dhamo et al., 2021; Zhai et al., 2023) utilize relation graphs to provide explicit control over object interactions, which are however too complicated and fussy for human users to specify. Moreover, previous works primarily represent objects by only categories (Luo et al., 2020; Paschalidou et al., 2021) or low-dimensional features (Wang et al., 2019; Tang et al., 2023) which lack visual appearance details, resulting in style inconsistency and constraining customization options in scene synthesis.

To address these issues, we present INSTRUCTSCENE, a novel generative framework for 3D indoor scene synthesis with natural language instructions. The overview of the proposed method is illustrated in Figure 1. INSTRUCTSCENE comprises two parts: a **semantic graph prior** and a **layout decoder**. In the first stage, it takes instructions about partial interior arrangement and object appearances, and learns the conditional distribution of semantic graphs for holistic scenes. In the second stage, harnessing the well-structured and informative graph latents, the layout decoder can easily embody scenes that exhibit semantic consistency while closely adhering to the provided instructions. With the learned semantic graph prior, INSTRUCTSCENE also achieves a wide range of instruction-driven generative tasks in a zero-shot manner.

---

[*]Corresponding author.

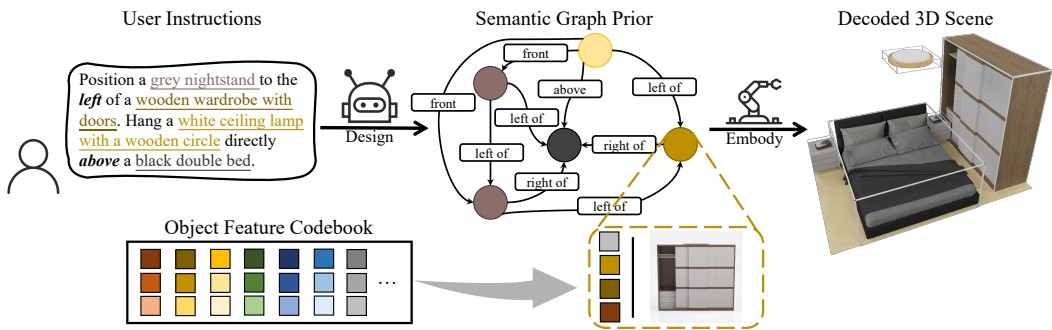

Figure 1: **Method overview.** (1) INSTRUCTSCENE first designs a holistic semantic graph based on user instruction. Within this graph, each node is an object endowed with semantic features and each edge represents a spatial relationship between objects. (2) It proceeds to place objects in a scene by decoding precise 7 degrees-of-freedom attributes for each object from the informative graph prior.

Specific conditional diffusion models are devised for both parts of INSTRUCTSCENE. Benefitting from the two-stage scheme, it can separately handle discrete and continuous attributes of indoor scenes, drastically reducing the burden of network optimization. To enhance the capability of aesthetic design, INSTRUCTSCENE also leverages object geometrics and appearances by quantizing semantic features from a multimodal-aligned model (Radford et al., 2021; Liu et al., 2023b).

To fit practical scenarios and promote the benchmarking of instruction-drive scene synthesis, we curate a high-quality dataset containing paired scenes and instructions with the help of large language and multimodal models (Li et al., 2022; Ouyang et al., 2022; OpenAI, 2023). Comprehensive quantitative evaluations reveal that INSTRUCTSCENE surpasses previous state-of-the-art methods by a large margin in terms of both generation controllability and fidelity. Each essential component of our method is carefully verified through ablation studies.

Our contributions can be summarized as follows:

- We present an instruction-driven generative framework that integrates a semantic graph prior and a layout decoder to improve the controllability and fidelity for 3D scene synthesis.
- The proposed general semantic graph prior jointly models appearance and layout distributions, facilitating various downstream applications in a zero-shot manner.
- We curate a high-quality dataset to promote the benchmarking of instruction-driven 3D scene synthesis, and quantitative experiments demonstrate that the proposed method significantly outperforms existing state-of-the-art techniques.

## 2 RELATED WORK

**Graph-driven 3D Scene Synthesis**   Graphs have been used to guide complex scene synthesis in the form of scene hierarchies (Li et al., 2019; Gao et al., 2023), parse trees (Purkait et al., 2020), scene graphs (Zhou et al., 2019a; Para et al., 2021), etc. Wang et al. (2019) utilize an image-based module and condition its outputs on the edges of a relation graph within each non-differentiable step. They also adopt an autoregressive model (Li et al., 2018) to generate relation graphs, which are however unconditional and with limited object attributes. Other works (Luo et al., 2020; Dhamo et al., 2021; Zhai et al., 2023) adopt conditional VAEs (Kingma & Welling, 2014; Sohn et al., 2015) with graph convolutional networks (Johnson et al., 2018) to generate layouts. While offering high controllability, these methods demand the specification of elaborate graph conditions, which are notably more intricate than those driven by natural languages.

**Language-driven 3D Scene Synthesis**   Early studies on language-driven scene synthesis are conducted through procedural modeling, resulting in a semi-automatic process (Chang et al., 2014; 2015a; 2017; Ma et al., 2018). With the advent of attention mechanisms (Vaswani et al., 2017), recent approaches (Wang et al., 2021; Paschalidou et al., 2021; Liu et al., 2023a; Tang et al., 2023) can implicitly acquire object relations by self-attention and condition scene synthesis with texts by cross-attention. However, text prompts in these works tend to be relatively simple, containing only object categories or lacking layout descriptions, limiting the expressiveness and customization. Implicit relation modeling also significantly hinders their controllability.

**Generative Models for Graphs**    There have been lots of endeavors on generative models for undirected graphs, molecules and scene graphs by autoregressive models (You et al., 2018; Garg et al., 2021), VAEs (Simonovsky & Komodakis, 2018; Verma et al., 2022), GANs (De Cao & Kipf, 2018; Martinkus et al., 2022) and diffusion models (Niu et al., 2020; Jo et al., 2022; Vignac et al., 2023; Kong et al., 2023). Longland et al. (2022) and Lo et al. (2023) employ VAE and GAN respectively for text-driven undirected simple graph generation without any semantics. In contrast, we present a pioneering effort to generate holistic semantic graphs with expressive instructions.

## 3    PRELIMINARY: DIFFUSION MODELS

Diffusion generative models (Sohl-Dickstein et al., 2015) consist of a non-parametric forward process and a learnable reverse process. The forward process progressively corrupts a data point from $q(\mathbf{x}_0)$ to a sequence of increasingly noisy latent variables: $q(\mathbf{x}_{1:T}|\mathbf{x}_0) = \prod_{t=1}^{T} q(\mathbf{x}_t|\mathbf{x}_{t-1})$. A neural network is trained to reverse the process by denoising them iteratively: $p_\psi(\mathbf{x}_{0:T}|\mathbf{c}) = p(\mathbf{x}_T) \prod_{t=1}^{T} p_\psi(\mathbf{x}_{t-1}|\mathbf{x}_t, \mathbf{c})$, where $\mathbf{c}$ is an optional condition to guide the reverse process as needed. These two processes are supposed to admit $p(\mathbf{x}_T) \approx q(\mathbf{x}_T|\mathbf{x}_0)$ for a sufficiently large $T$. The generative model is optimized by minimizing a variational upper bound on $\mathbb{E}_{q(\mathbf{x}_0)}[-\log p_\psi(\mathbf{x}_0)]$:

$$\mathcal{L}_{\text{vb}} := \mathbb{E}_{q(\mathbf{x}_0)}[\ \underbrace{D_{\text{KL}}[q(\mathbf{x}_T|\mathbf{x}_0)\|p(\mathbf{x}_T)]}_{\mathcal{L}_T} + \sum_{t=2}^{T} \mathcal{L}_{t-1} \underbrace{-\mathbb{E}_{q(\mathbf{x}_1|\mathbf{x}_0)}[\log p_\psi(\mathbf{x}_0|\mathbf{x}_1, \mathbf{c})]}_{\mathcal{L}_0}\ ], \qquad (1)$$

where $\mathcal{L}_{t-1} := D_{\text{KL}}[q(\mathbf{x}_{t-1}|\mathbf{x}_t, \mathbf{x}_0)\|p_\psi(\mathbf{x}_{t-1}|\mathbf{x}_t, \mathbf{c})]$ and $\mathcal{L}_T$ is constant during training so can be ignored. $D_{\text{KL}}[\cdot\|\cdot]$ indicates the KL divergence between two distributions.

## 4    METHOD

### 4.1    PROBLEM STATEMENT

Denote $\mathcal{S} := \{\mathcal{S}_1, \ldots, \mathcal{S}_M\}$ as a collection of indoor scenes. Each scene $\mathcal{S}_i$ is composed of multiple objects $\mathcal{O}_i := \{\mathbf{o}_j^i\}_{j=1}^{N_i}$ with distinct attributes $\mathbf{o}_j^i := \{c_j^i, \mathbf{t}_j^i, \mathbf{s}_j^i, r_j^i, \mathbf{f}_j^i\}$, including category $c_j^i \in \{1, ..., K_c\}$, where $K_c$ is the number of object classes in $\mathcal{S}$, location $\mathbf{t}_j^i \in \mathbb{R}^3$, axis-aligned size $\mathbf{s}_j^i \in \mathbb{R}^3$, orientation $r_j^i \in \mathbb{R}$ and semantic feature $\mathbf{f}_j^i \in \mathbb{R}^d$, where $d$ is the dimension of the feature vector. To set up a 3D scene, one can generate each 3D object or retrieve it from a database by $c$ and $\mathbf{f}$. They are then resized and transformed to the same scene coordinate by corresponding $\mathbf{t}$, $\mathbf{s}$ and $r$. To simplify the process, we opt to retrieve 3D objects from a high-quality dataset, and leave the generative models of each object conditioned on $c$ and $\mathbf{f}$ for future work.

Given instructions $\mathbf{y}$, our goal is to learn the conditional scene distribution $q(\mathcal{S}|\mathbf{y})$. Rather than direct modeling (Paschalidou et al., 2021; Tang et al., 2023), we employ well-structured and informative graphs to serve as general and semantic latents. Each graph $\mathcal{G}_i$ contains a node set $\mathcal{V}_i := \{\mathbf{v}_j^i\}_{j=1}^{N_i}$ and a directed edge set $\mathcal{E}_i := \{e_{jk}^i|\mathbf{v}_j^i, \mathbf{v}_k^i \in \mathcal{V}_i\}$. A node $\mathbf{v}_j^i$ functions as a high-level representation of an object $\mathbf{o}_j^i$, and a directed edge $e_{jk}^i$ explicitly conveys the relations between objects.

To this end, we propose a generative framework, INSTRUCTSCENE, that consists of two components: (1) **semantic graph prior** $p_\phi(\mathcal{G}|\mathbf{y})$ (Sec. 4.2) that jointly models high-level object and relation distributions conditioned on $\mathbf{y}$; (2) **layout decoder** $p_\theta(\mathcal{S}|\mathcal{G})$ (Sec. 4.3) that produces precise layout configurations with semantic graphs. Since $\mathcal{G}$ is deterministic by corresponding $\mathcal{S}$, the two networks together yield an instruction-driven generative model for 3D indoor scenes:

$$p_{\phi,\theta}(\mathcal{S}|\mathbf{y}) = p_{\phi,\theta}(\mathcal{S}, \mathcal{G}|\mathbf{y}) = p_\phi(\mathcal{G}|\mathbf{y})p_\theta(\mathcal{S}|\mathcal{G}). \qquad (2)$$

### 4.2    SEMANTIC GRAPH PRIOR

The spatial relations are defined based on distances and relative orientations, such as "left", "closely in front of", "above" and "too far away (none)". Details about the definitions are provided in Appendeix A.1. Layout configurations including $\mathbf{t}$, $\mathbf{s}$ and $r$ can be derived from spatial relations, so we leave them to the decoder $p_\theta(\mathcal{S}|\mathcal{G})$. Denote $\mathbf{v} := \{c, \mathbf{f}\}$ and $e \in \{1, \ldots, K_e\}$, where $K_e$ is the number of relation classes.

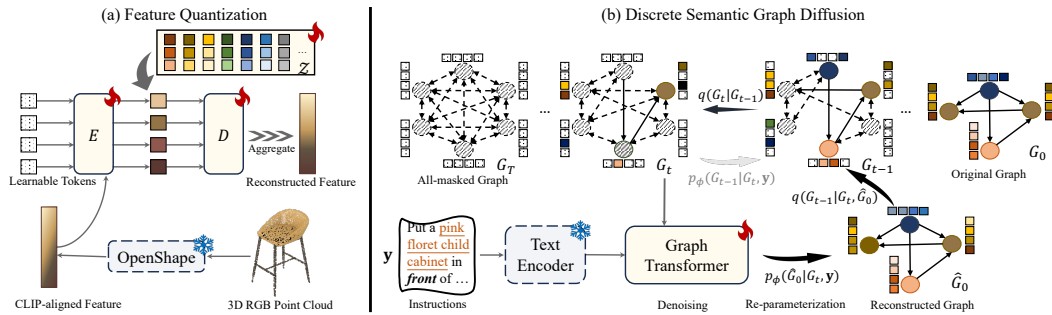

Figure 2: **Semantic Graph Prior**. (a) **Feature Quantization**. Semantic features for 3D objects are extracted from a frozen multimodal-aligned point cloud encoder and then quantized by codebook entries. (b) **Discrete Semantic Graph Diffusion**. Three categorical variables in $G_0$ are independently diffused; Empty states are not depicted for concision; A graph Transformer with a frozen text encoder learns the semantic graph prior by iteratively denoising corrupted graphs.

### 4.2.1 FEATURE QUANTIZATION

High-dimensional features, such as those with $d = 1280$ in OpenCLIP `ViT-bigG/14` (Liu et al., 2023b), are too complicated to model. We circumvent this drawback by introducing a vector-quantized variational autoencoder for feature vectors, coined as $f$**VQ-VAE**. The intuition behind it is that there are general intrinsic characteristics shared among objects, encompassing attributes like colors, materials and basic geometric shapes. Indexing semantic features from a codebook could dramatically reduce the cost of operating in a continuous space.

Formally, $f$VQ-VAE contains a pair of encoder $E$ and decoder $D$, along with a codebook $\mathcal{Z} \in \mathbb{R}^{K_f \times d_{\mathcal{Z}}}$, where $K_f$ and $d_{\mathcal{Z}}$ are its size and dimension respectively. To concurrently capture object visual appearances and geometric shapes, we employ a multimodal-aligned point cloud encoder, OpenShape (Liu et al., 2023b), to extract object semantic features. The diagram for $f$VQ-VAE is presented in Figure 2(a). It is trained to maximize the evidence lower bound (ELBO) for $\log p(\mathbf{f})$:

$$\mathbb{E}_{\mathbf{z} \sim p_E(\mathbf{z}|\mathbf{f})} \left[ \log p_D(\mathbf{f}|\mathbf{z}) - \beta D_{\mathrm{KL}}(p_E(\mathbf{z}|\mathbf{f}) \| p(\mathbf{z})) \right], \tag{3}$$

where $\mathbf{z} \in \mathbb{R}^{n_f \times d_{\mathcal{Z}}}$ consists of $n_f$ vectors indexed by a sequence of scalars $f := [f_m]_{m=1}^{n_f}$, where each scalar $f_m \in \{1, \ldots, K_f\}$. Since the quantization operation is non-differentiable, gumbel-softmax relaxation (Jang et al., 2016; Ramesh et al., 2021) is adopted to optimize the ELBO.

### 4.2.2 DISCRETE SEMANTIC GRAPH DIFFUSION

After the feature quantization, all attributes in a semantic graph are categorical, $\mathcal{G}_i := (\mathcal{C}_i, \mathcal{F}_i, \mathcal{E}_i)$, where $\mathcal{C}_i := \{1, \ldots, K_c\}^{N_i}$, $\mathcal{E}_i := \{1, \ldots, K_e\}^{N_i \times N_i}$ and $\mathcal{F}_i := \{1, \ldots, K_f\}^{N_i \times n_f}$. While it is possible to embed discrete variables in continuous spaces using one-hot encodings, it diminishes the sparsity inherent in the original data and imposes a substantial burden on network optimization. Instead, we propose to model the semantic graph prior through discrete diffusion models.

For a scalar discrete random variable with $K$ categories $x \in \{1, \ldots, K\}$, diffusion noise is defined by a series of transition matrices $\mathbf{Q} \in \mathbb{R}^{K \times K}$. The forward process at timestep $t$ is expressed as $q(x_t|x_{t-1}) := \mathbf{x}_t^\top \mathbf{Q}_t \mathbf{x}_{t-1}$, where $\mathbf{x}_t \in \mathbb{R}^K$ is the column one-hot encoding for $x_t$ and $[\mathbf{Q}_t]_{mn} := q(x_t = m|x_{t-1} = n)$ is the probability that $x_{t-1}$ transits to the category $m$ from $n$. The probabilistic distribution of $x_t$ can be directly derived from $x_0$: $q(x_t|x_0) := \mathbf{x}_t^\top \bar{\mathbf{Q}}_t \mathbf{x}_0$, where $\bar{\mathbf{Q}}_t := \mathbf{Q}_t \cdots \mathbf{Q}_1$.

Instead of commonly used Gaussian or uniform transitions for graph generation (Niu et al., 2020; Hoogeboom et al., 2021; Jo et al., 2022; Vignac et al., 2023), we propose to diffuse semantic graphs by independently masking graph attributes (i.e., object class $c$, quantized feature indices $f$ and relation $e$) by introducing an absorbing state `[MASK]` (Austin et al., 2021; Gu et al., 2022) to each uniform transition matrix. For object class $c$, its transition matrix is defined as:

$$\mathbf{Q}_t^{\mathbf{C}} := \begin{bmatrix} \alpha_t^c + \beta_t^c & \beta_t^c & \cdots & \beta_t^c & 0 \\ \beta_t^c & \alpha_t^c + \beta_t^c & \cdots & \beta_t^c & 0 \\ \vdots & \vdots & \ddots & \beta_t^c & 0 \\ \beta_t^c & \beta_t^c & \beta_t^c & \alpha_t^c + \beta_t^c & 0 \\ \gamma_t^c & \gamma_t^c & \gamma_t^c & \gamma_t^c & 1 \end{bmatrix}, \tag{4}$$

by which $c_t$ has a probability of $\gamma_t^c$ to be masked, a probability of $\alpha_t^c$ to maintain the same, leaving a chance of $1 - \gamma_t^c - \alpha_t^c$ for uniform sampling. `[MASK]` will always stay in its own state. Transition matrices for $f$ and $e$, denoted as $\mathbf{Q}_t^{\mathsf{F}}$ and $\mathbf{Q}_t^{\mathsf{E}}$ respectively, exhibit analogous structures. Schedules of $(\alpha_t, \beta_t, \gamma_t)$ are designed to admit that the initial states for semantic graphs are all masked.

Since the number of objects varies across different scenes, semantic graphs are padded by empty states to maintain a consistent number of $N$ objects. One-hot encodings for scalar variables $c$, $f$ and $e$ in a scene are denoted as $\mathbf{C} \in \mathbb{R}^{N \times (K_c+2)}$, $\mathbf{F} \in \mathbb{R}^{N \times n_f \times (K_f+2)}$ and $\mathbf{E} \in \mathbb{R}^{N \times N \times (K_e+2)}$ respectively. Here "+2" accounts for the two extra states (i.e., empty state and mask state) for each variable. A one-hot encoded semantic graph $G_0 := (\mathbf{C}_0, \mathbf{F}_0, \mathbf{E}_0)$ at timestep $t$ is formulated as

$$q(G_t|G_0) = (\bar{\mathbf{Q}}_t^{\mathsf{C}} \mathbf{C}_0, \bar{\mathbf{Q}}_t^{\mathsf{F}} \mathbf{F}_0, \bar{\mathbf{Q}}_t^{\mathsf{E}} \mathbf{E}_0). \tag{5}$$

The process for learning the graph prior is illustrated in Figure 2(b). The independent diffusion with mask states offers two significant advantages:

- Perturbed states for one variable (e.g., $\mathbf{C}$) could be recovered by incorporating information from uncorrupted portions of the other variables (e.g., $\mathbf{F}$ and $\mathbf{E}$), compelling the semantic graph prior to learning from the interactions among different scene attributes.

- The introduction of mask states facilitates the distinction between corrupted states and clean ones, thus simplifying the denoising task.

These benefits are critical especially for intricate semantic graphs and diverse downstream generative tasks, compared with simple graph generative tasks (Niu et al., 2020; Jo et al., 2022; Vignac et al., 2023). Ablation study on the choice of $\mathbf{Q}$ is provided in Sec. 5.5.2.

Output of the graph prior network is re-parameterized to produce the clean scene graphs $\hat{G}_0$, which is then diffused to get the predicted posterior for computing the variational bound in Equation 1: $p_\phi(G_{t-1}|G_t, \mathbf{y}) \propto \sum_{\hat{G}_0} q(G_{t-1}|G_t, \hat{G}_0) p_\phi(\hat{G}_0|G_t, \mathbf{y})$. Training objective for $p_\phi$ is a weighted summation of variational bounds for three random variables conditioned on $\mathbf{y}$:

$$\mathcal{L}_{\mathrm{vb}}^{\mathcal{G}|\mathbf{y}} := \mathcal{L}_{\mathrm{vb}}^{\mathbf{C}|\mathbf{y}} + \lambda_f \cdot \mathcal{L}_{\mathrm{vb}}^{\mathbf{F}|\mathbf{y}} + \lambda_e \cdot \mathcal{L}_{\mathrm{vb}}^{\mathbf{E}|\mathbf{y}}, \tag{6}$$

where $\lambda_e, \lambda_f \in \mathbb{R}^+$ are hyperparameters to adjust the relative importance of three components in the semantic graph.

### 4.3 3D LAYOUT DECODER

Instantiating 3D scenes becomes easy with semantic graph prior. Denote layout configurations of $\mathcal{S}_i$ as $\{\mathbf{l}_j^i\}_{j=1}^{N_i}$, where $\mathbf{l}_j^i := \mathbf{o}_j^i - \mathbf{v}_j^i = \{\mathbf{t}_j^i, \mathbf{s}_j^i, r_j^i\}$. $SO(2)$ rotation is parameterize by $[\cos(r), \sin(r)]^\top$ to continuously represent $r$ (Zhou et al., 2019b). Consequently, the layout of $\mathcal{S}_i$ can be expressed as 2D matrices $\mathbf{L}_i \in \mathbb{R}^{N_i \times 8}$. Note that $\mathcal{S} = (\mathbf{L}, \mathcal{G})$, so generating indoor scenes $p_\theta(\mathcal{S}|\mathcal{G})$ is equivalent to learning the conditional distributions of layout configurations $p_\theta(\mathbf{L}|\mathcal{G})$.

A diffusion model with variance-preserving Gaussian kernels (Ho et al., 2020; Song et al., 2020) is adopted to learn $p_\theta(\mathbf{L}|\mathcal{G})$. Its forward process is $q(\mathbf{L}_t|\mathbf{L}_0) := \mathcal{N}(\mathbf{L}_t; \sqrt{\bar{\alpha}_t}\mathbf{L}_0, (1 - \bar{\alpha}_t)\mathbf{I})$. The reverse process is modeled as $p_\theta(\mathbf{L}_{t-1}|\mathbf{L}_t, \mathcal{G}) := \mathcal{N}(\mathbf{L}_{t-1}; \boldsymbol{\mu}_\theta(\mathbf{L}_t, t, \mathcal{G}); \boldsymbol{\Sigma}_\theta(\mathbf{L}_t, t, \mathcal{G}))$. Following Ho et al. (2020), the variational bound in Equation 1 for the decoder $p_\theta(\mathbf{L}|\mathcal{G})$ is reweighted and simplified:

$$\begin{aligned} \mathcal{L}_{\mathrm{simple}} &:= \mathbb{E}_{\mathbf{L}_0, t, \boldsymbol{\epsilon}} \left[ \|\boldsymbol{\epsilon} - \boldsymbol{\epsilon}_\theta(\mathbf{L}_t, t, \mathcal{G})\|^2 \right] \\ &= \mathbb{E}_{\mathbf{L}_0, t, \boldsymbol{\epsilon}} \left[ \|\boldsymbol{\epsilon} - \boldsymbol{\epsilon}_\theta(\sqrt{\bar{\alpha}_t}\mathbf{L}_0 + \sqrt{1 - \bar{\alpha}_t}\boldsymbol{\epsilon}, t, \mathcal{G})\|^2 \right], \end{aligned} \tag{7}$$

where $t$ is sampled from a uniform distribution $\mathcal{U}(1, T)$ and $\boldsymbol{\epsilon}$ is sampled from a standard normal distribution $\mathcal{N}(\mathbf{0}, \mathbf{I})$. Diagram of the layout decoder is depicted in Figure 3(a). Intuitively, the network is trained to predict noise $\boldsymbol{\epsilon}$ in the corrupted data $\mathbf{L}_t$.

### 4.4 MODEL ARCHITECTURE

We use the general-purpose Transformer (Vaswani et al., 2017) for all models across tasks.

Figure 3: (a) **3D Layout Decoder**. Gaussian noises are sampled to attach at every node of semantic graphs; A graph Transformer processes these graphs iteratively to remove noises and generate layout configurations. (b) **Graph Transformer**. A graph Transformer consists of a stack of $M$ blocks, each comprising graph attention, MLP and optional cross-attention modules; AdaLN and multi-head scheme are not depicted for concision.

**Vanilla Transformer**    As illustrated in Figure 2(a), $n_f$ learnable tokens are employed with a stack of cross-attentions to extract information from object features $\mathbf{f}$ in the encoder $E$ in $f$VQ-VAE. Regarding the decoder $D$, $n_f$ vectors retrieved from the codebook $\mathcal{Z}$ are fed to another Transformer, and an average pooling on the top of it is applied to aggregate information.

**Graph Transformer**    The prior and decoder share the same model architecture as shown in Figure 3(b). Since relation $e_{jk}$ can be determined by $e_{kj}$, only the upper triangular part of the relation matrix is necessary. Object categories and features together form input tokens for Transformers. Message passing on graphs is operated via node self-attention and node-edge fusion with FiLM (Perez et al., 2018), which linearly modulates edge embeddings and node attention matrices before softmax (Dwivedi & Bresson, 2021; Vignac et al., 2023). Timestep for diffusion $t$ is injected by AdaLN (Ba et al., 2016; Dhariwal & Nichol, 2021). In the prior $p_\phi(\mathcal{G}|\mathbf{y})$, instructions are embedded by a frozen text encoder and consistently influence network outputs by cross-attention mechanisms. Layout decoder $p_\theta(\mathcal{S}|\mathcal{G})$ is conditioned on semantic graphs by appending sampled Gaussian noises on node embeddings, which are then iteratively denoised to produce layout attributes.

**Permutation Non-invariance**    Although $\mathcal{G}$ should ideally remain invariant to node permutations, invariant diffusion models could encounter learning challenges for multi-mode modeling. Thus, each node feature is added with positional encodings (Vaswani et al., 2017; Tang et al., 2023; Lei et al., 2023) before the permutation-equivariant Transformer. Exchangeability for graph prior distributions is strived by random permutation augmentation during the training process. Ablation on the permutation non-invariance is provided in Sec. 5.5.2.

## 5  EXPERIMENTS

### 5.1  SCENE-INSTRUCTION PAIR DATASET

All experiments are conducted on 3D-FRONT (Fu et al., 2021a), a professionally designed collection of synthetic indoor scenes. However, it does not contain any descriptions of room layouts or object appearances. To construct a high-quality scene-instruction paired dataset, we initially extract view-dependent spatial relations with predefined rules. The dataset is further enhanced by captioning objects with BLIP (Li et al., 2022). To ensure the accuracy of descriptions, the generated captions are filtered by ChatGPT (Ouyang et al., 2022; OpenAI, 2023) with object ground-truth categories. The final instructions are derived from randomly selected relation triplets. Details on dataset curation can be found in Appendix A.

### 5.2  EXPERIMENTAL SETTINGS

**Baselines**    We compare our method with two state-of-the-art approaches for 3D scene generative tasks: (1) ATISS (Paschalidou et al., 2021), a Transformer-based auto-regressive network that regards scenes as sets of unordered objects, and generates objects and their attributes sequentially. (2) DiffuScene (Tang et al., 2023), a diffusion model with Gaussian kernels that treats object attributes in one scene as a 2D matrix after padding them to a fixed size. Both of these methods can be conditioned on text prompts by cross-attention with a pretrained text encoder. Our preliminary

Table 1: Quantitive evaluations for instruction-driven synthesis by ATISS (Paschalidou et al., 2021), DiffuScene (Tang et al., 2023) and our method on three room types. Higher iRecall, lower FID, FID$^{\text{CLIP}}$ and KID indicate better synthesis quality. For SCA, a score closer to 50% is better. Standard deviation values are provided as subscripts.

| Instruction-driven Synthesis | | $\uparrow$ iRecall$_\%$ | $\downarrow$ FID | $\downarrow$ FID$^{\text{CLIP}}$ | $\downarrow$ KID$_{\times 1e\text{-}3}$ | SCA$_\%$ |
|---|---|---|---|---|---|---|
| | ATISS | $48.13_{\pm 2.50}$ | $119.73_{\pm 1.55}$ | $6.95_{\pm 0.06}$ | $0.39_{\pm 0.02}$ | $59.17_{\pm 1.39}$ |
| Bedroom | DiffuScene | $56.43_{\pm 2.07}$ | $123.09_{\pm 0.79}$ | $7.13_{\pm 0.16}$ | $0.39_{\pm 0.01}$ | $60.49_{\pm 2.96}$ |
| | Ours | $\mathbf{73.64}_{\pm 1.37}$ | $\mathbf{114.78}_{\pm 1.19}$ | $\mathbf{6.65}_{\pm 0.18}$ | $\mathbf{0.32}_{\pm 0.03}$ | $\mathbf{56.02}_{\pm 1.43}$ |
| | ATISS | $29.50_{\pm 3.67}$ | $117.67_{\pm 2.32}$ | $6.08_{\pm 0.13}$ | $17.60_{\pm 2.65}$ | $69.38_{\pm 3.38}$ |
| Living room | DiffuScene | $31.15_{\pm 2.49}$ | $122.20_{\pm 1.09}$ | $6.10_{\pm 0.11}$ | $16.49_{\pm 1.24}$ | $72.92_{\pm 1.29}$ |
| | Ours | $\mathbf{56.81}_{\pm 2.85}$ | $\mathbf{110.39}_{\pm 0.78}$ | $\mathbf{5.37}_{\pm 0.07}$ | $\mathbf{8.16}_{\pm 0.56}$ | $\mathbf{65.42}_{\pm 2.52}$ |
| | ATISS | $37.58_{\pm 1.99}$ | $137.10_{\pm 0.34}$ | $8.49_{\pm 0.23}$ | $23.60_{\pm 2.52}$ | $67.61_{\pm 3.23}$ |
| Dining room | DiffuScene | $37.87_{\pm 2.76}$ | $145.48_{\pm 1.36}$ | $8.63_{\pm 0.31}$ | $24.08_{\pm 1.90}$ | $70.57_{\pm 2.14}$ |
| | Ours | $\mathbf{61.23}_{\pm 1.67}$ | $\mathbf{129.76}_{\pm 1.61}$ | $\mathbf{7.67}_{\pm 0.18}$ | $\mathbf{13.24}_{\pm 1.79}$ | $\mathbf{64.20}_{\pm 1.90}$ |

experiments suggest that both baselines encounter difficulties in modeling high-dimensional semantic feature distributions, consequently impacting their performance in generating other attributes. Therefore, we augment them to generate quantized features. Further implementation details about baselines and our method are provided in Appendix B.1 and B.2.

**Evaluation Metrics**  To assess the controllability of layouts, we use a metric named "instruction recall" (iRecall), which quantifies the proportion of the required triplets "(subject, relation, object)" occurring in synthesized scenes to all provided in instructions. It is a stringent metric that takes into account all three elements in a layout relation simultaneously. Following previous works (Paschalidou et al., 2021; Liu et al., 2023a; Tang et al., 2023), we also report Fréchet Inception Distance (FID) (Heusel et al., 2017), FID$^{\text{CLIP}}$ (Kynkäänniemi et al., 2022), which computes FID scores by CLIP features (Radford et al., 2021), Kernel Inception Distance (KID) (Bińkowski et al., 2018), scene classification accuracy (SCA). These metrics evaluate the overall quality of synthesized scenes and rely on rendered images. We use Blender (Community, 2018) to produce high-quality images for both synthesized and real scenes. For more details on evaluation, please refer to Appendix B.3.

## 5.3 INSTRUCTION-DRIVEN SCENE SYNTHESIS

Table 1 presents the quantitive evaluations for synthesizing 3D scenes with instructions. We report the average scores of five runs with different random seeds. As demonstrated, even with the enhancement of quantized semantic features, two baseline methods continue to demonstrate inferior performance compared to ours. ATISS outperforms DiffuScene in terms of generation fidelity, owing to its capacity to model in discrete spaces. DiffuScene shows better controllability to ATISS because it affords global visibility of samples during generation. Our proposed INSTRUCTSCENE exhibits the best of both worlds. Remarkably, we achieve a substantial advancement in controllability, measured in iRecall, for scene generative models, surpassing current state-of-the-art approaches by about **15%~25%** across various room types, all while maintaining high fidelity. It is noteworthy that INSTRUCTSCENE excels in handling more complex scenes, such as living and dining rooms, which typically comprise an average of 20 objects, in contrast to bedrooms, which have only 8 objects on average, revealing the benefits of modeling intricate 3D scenes associated with the semantic graph prior. Qualitative visualizations are provided in Appendix C.1.

## 5.4 ZERO-SHOT APPLICATIONS

Thanks to the discrete design and mask modeling, the learned semantic graph prior is capable of diverse downstream tasks without any fine-tuning. We investigate four zero-shot tasks, including stylization, re-arrangement, completion, and unconditional generation. The first three tasks can be regarded as conditional synthesis guided by both instructions and partial scene attributes.

Stylization and re-arrangement task can be formulated as $p_\phi(\mathbf{f}|c, \mathbf{t}, \mathbf{s}, r, \mathbf{y})$ and $p_{\phi,\theta}(\mathbf{t}, \mathbf{s}, r|c, \mathbf{f}, \mathbf{y})$ respectively. In the completion task, we intend to add new objects $\{\mathbf{o}_k^i\}$ to a partial scene $\mathcal{S}_i$ with

Table 2: Quantitive evaluations for zero-shot generative applications on three room types. "Uncond." stands for unconditional scene synthesis.

| Zero-shot Applications | | Stylization | | Re-arrangement | | Completion | | Uncond. |
|---|---|---|---|---|---|---|---|---|
| | | $\uparrow \Delta_{\times 1e-3}$ | $\downarrow$ FID | $\uparrow$ iRecall$_\%$ | $\downarrow$ FID | $\uparrow$ iRecall$_\%$ | $\downarrow$ FID | $\downarrow$ FID |
| Bedroom | ATISS | 3.44 | 123.91 | 61.22 | 107.67 | 64.90 | 89.77 | 134.51 |
| | DiffuScene | 1.08 | 127.35 | 68.57 | 106.15 | 48.57 | 96.28 | 135.46 |
| | Ours | **6.34** | **122.73** | **79.59** | **105.27** | **69.80** | **82.98** | **124.97** |
| Living room | ATISS | -3.57 | 110.85 | 31.97 | 117.97 | 43.20 | 106.48 | 129.23 |
| | DiffuScene | -2.69 | 112.80 | 41.50 | 115.30 | 19.73 | 95.94 | 129.75 |
| | Ours | **0.28** | **109.39** | **56.12** | **106.85** | **46.94** | **92.52** | **117.62** |
| Dining room | ATISS | -1.11 | 131.14 | 36.06 | 134.54 | 57.99 | 122.44 | 147.52 |
| | DiffuScene | -2.98 | 135.20 | 46.84 | 133.73 | 32.34 | 115.08 | 150.81 |
| | Ours | **1.69** | **128.78** | **62.08** | **125.07** | **60.59** | **107.86** | **137.52** |

instructions. By filling the partial scene attributes with `[MASK]` tokens, we treat them as intermediate states during discrete graph denoising, allowing for a straightforward adaptation of the learned semantic graph prior to these tasks in a zero-shot manner. Unconditional synthesis is implemented by simply setting text features as zeros. To assess controllability in the stylization task, we define $\Delta := \frac{1}{N} \sum_{i=1}^{N} \text{CosSim}(\mathbf{f}_i, \mathbf{d}_i^{\text{style}}) - \text{CosSim}(\mathbf{f}_i, \mathbf{d}_i^{\text{class}})$, where $\mathbf{d}_i^{\text{style}}$ represents the CLIP text feature of object class name with the desired style, and $\mathbf{d}_i^{\text{class}}$ is the CLIP text feature with only class information. $\text{CosSim}(\cdot, \cdot)$ calculates the cosine similarity between two vectors.

Evaluations on zero-shot applications are reported in Table 2. Our method consistently outperforms two strong baselines in both controllability and fidelity. While ATISS, as an auto-regressive model, is a natural fit for the completion task, its unidirectional dependency chain limits its effectiveness for tasks requiring global scene modeling, such as re-arrangement. DiffuScene can adapt to these tasks by replacing the known parts with the noised corresponding scene attributes during sampling, similar to image in-painting (Meng et al., 2021; Nichol et al., 2022). However, the known attributes are greatly corrupted in the early steps, which could misguide the denoising direction, and therefore necessitate fine-tuning. Additionally, DiffuScene also faces challenges in searching for semantic features in a continuous space for stylization. In contrast, INSTRUCTSCENE globally models scene attributes and treats partial scene attributes as intermediate discrete states during training. These designs effectively eliminate the training-test gap, rendering it highly versatile for a wide range of downstream tasks. Visualizations of zero-shot applications are available in Appendix C.2.

## 5.5 ABLATION STUDIES

### 5.5.1 CONFIGURATIONS FOR DIFFUSION MODELS

**Diffusion Timesteps** Although containing two diffusion models, our method could achieve better efficiency by reducing the steps of reverse processes without a noticeable decline in performance. This stems from the fact that each stage in INSTRUCTSCENE tackles an easier denoising task compared to the single-stage DiffuScene. Following the original setting of Tang et al. (2023), the timestep value ($T$) for DiffuScene is set to `1000`. While for INSTRUCTSCENE, we find $T = 100$ and `10` is sufficient for $p_\phi(\mathcal{G}|\mathbf{y})$ and $p_\theta(\mathcal{S}|\mathcal{G})$ respectively. Evaluation results with different timesteps are presented in Figure 4(a), with values averaged on three room types. The plotted timesteps for our method are "100+1000", "100+400", "100+100", "100+10", "50+10" and "25+10", where the first number represents $T$ for the prior and the latter is for the decoder.

**Classifier-Free Guidance** Classifier-free guidance (CFG) (Ho & Salimans, 2021) is a widely used technique to trade off controllability with diversity. We do not adopt it in previous experiments for a fair comparison, as the sequential attribute decoding hinders ATISS from realizing the benefits offered by CFG. To assess its effectiveness for diffusion models, we randomly remove instruction conditions on `20%` of samples during training, inducing an unconditional generation. At inference, CFG is implemented by adjusting conditional log-likelihoods away from unconditional counterparts:

$$\tilde{p}_\phi(\hat{G}_0|G_t, \mathbf{y}) := (1 + s) \cdot p_\phi(\hat{G}_0|G_t, \mathbf{y}) - s \cdot p_\phi(\hat{G}_0|G_t), \qquad (8)$$

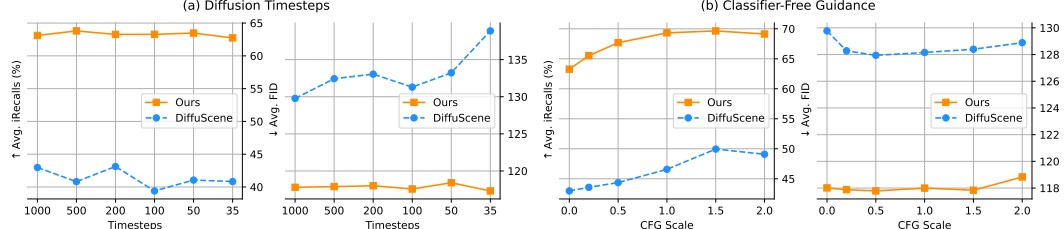

Figure 4: Ablation studies on configurations for diffusion models, including diffusion timesteps and classifier-free guidance scales.

Table 3: Ablation studies on different strategies to learn semantic graph prior $p_\phi(\mathcal{G}|\mathbf{y})$. "Perm. Invar." means permutation-invariant graph modeling.

| Graph Prior | Ours | Gaussian | Joint Mask | Uniform | Perm. Invar. |
|---|---|---|---|---|---|
| ↑ iRecall$_\%$ | **73.64**$_{\pm 1.37}$ | 34.18$_{\pm 2.53}$ | 34.21$_{\pm 2.79}$ | 69.22$_{\pm 3.25}$ | 70.49$_{\pm 2.50}$ |
| ↓ FID | **114.78**$_{\pm 1.19}$ | 128.98$_{\pm 0.97}$ | 130.86$_{\pm 2.76}$ | 139.61$_{\pm 1.06}$ | 116.53$_{\pm 1.35}$ |
| ↓ FID$^{\mathrm{CLIP}}$ | **6.65**$_{\pm 0.18}$ | 7.30$_{\pm 0.03}$ | 7.59$_{\pm 0.17}$ | 8.82$_{\pm 0.24}$ | 6.69$_{\pm 0.16}$ |
| ↓ KID$_{\times 1e\text{-}3}$ | **0.32**$_{\pm 0.03}$ | 2.63$_{\pm 0.73}$ | 4.82$_{\pm 1.69}$ | 10.55$_{\pm 1.19}$ | 0.37$_{\pm 0.02}$ |
| SCA$_\%$ | **56.02**$_{\pm 0.91}$ | 57.10$_{\pm 3.22}$ | 60.37$_{\pm 3.13}$ | 76.79$_{\pm 3.14}$ | 58.64$_{\pm 1.33}$ |

where $s$ is a hyperparameter to control the scale of CFG. Performance for diffusion-based models with different CFG scales are plotted in Figure 4(b), where values are averaged over three room types. Within an appropriate range of scales, CFG can effectively enhance the controllability for instructive-driven 3D scene synthesis, while large scales can lead to a performance decline. Though DiffuScene also benefits from CFG, our method still significantly outperforms it in both metrics.

### 5.5.2 LEARNING SEMANTIC GRAPH PRIOR

We explore different strategies to learn the proposed semantic graph prior. All experiments are conducted on the bedroom dataset. Quantitative results are presented in Table 3.

**Transition Matrices for Learning Graph Prior**    We investigate the effects of different transition matrices for learning the proposed semantic graph prior, including: (1) Embed all categorical variables into their one-hot encodings and diffuse them by Gaussian kernels, which is similar to Niu et al. (2020) and Jo et al. (2022); (2) Jointly masking **F** and **E** along with nodes **C** in a graph, so only the attributes of other objects can be utilized for recovery; (3) Adopt uniform transition matrices without mask states, which is similar to Vignac et al. (2023). Evaluations on both controllability and fidelity reveal the advantages of our independent mask strategy.

**Permutation Non-invariance**    Unlike previous studies on graph generation (Niu et al., 2020; Jo et al., 2022; Vignac et al., 2023), we depart from the convention of permutation-invariant modeling to ease the learning process of semantic graph prior. We strive to preserve exchangeable graph distributions by randomly shuffling object orders during training. Performance for invariant graph prior is provided in the last column of Table 3. Its performance declines due to the unnecessary imposition of invariance in scene synthesis.

## 6 CONCLUSION

By integrating a semantic graph prior and a layout decoder, we propose a novel generative framework, INSTRUCTSCENE, that significantly improves the controllability and fidelity of 3D indoor scene synthesis, providing a user-friendly interface through instructions in natural languages. Benefits from the design of semantic graph prior, our method can also apply to diverse applications without any fine-tuning. The controllability and versatility positions INSTRUCTSCENE as a promising tool. We hope this work could help in practical scenarios, such as facilitating interior design, delivering immersive metaverse experiences, simulations for embodied agents, developing cutting-edge VR/AR applications, etc. We discuss the limitations of our method and future work in Appendix D.

## ETHICS STATEMENT

Several large pretrained models are incorporated in this work, including OpenShape (Liu et al., 2023b) for object semantic feature extraction, CLIP (Radford et al., 2021) for text feature extraction, BLIP (Li et al., 2022) for object captioning and ChatGPT (Ouyang et al., 2022; OpenAI, 2023) for caption refinement. Most of these models are trained on large-scale datasets collected from the web, lacking rigorous content filtering, thereby potentially encompassing harmful material. We curate the dataset and train our method based on these models, thus may inherit these imperfections. Given that our generative framework is trained only on indoor scene datasets, it exhibits less probability of propagating negative consequences compared to the synthesis and editing methods on human faces and natural images. Nevertheless, we will still explicitly specify permissible applications of our system through appropriate licensing to mitigate potential adverse societal impacts.

## REPRODUCIBILITY STATEMENT

To ensure the reproducibility of our method, we include the details of dataset processing in Appendex A, including the rule-based spatial relation definitions (A.1) and the used prompt and hyperparameters for ChatGPT to refine object descriptions (A.2). Implementation details are also provided in Appdex B, including baseline reproductions (B.1), model hyperparameter disclosure (B.2) and evaluation metric computations (B.3). Our instruction-scene pair dataset and code for both training and evaluation can be found in `https://chenguolin.github.io/projects/InstructScene`.

## ACKNOWLAGEMENT

This work is supported by National Key R&D Program of China (2022ZD0160305).

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

# A    DATASET PREPARATION

Following previous works (Paschalidou et al., 2021; Tang et al., 2023; Liu et al., 2023a), we use three types of indoor rooms in 3D-FRONT (Fu et al., 2021a) and preprocess the dataset by filtering some problematic samples, resulting in 4041 bedrooms, 813 living rooms and 900 dining rooms. The number of objects $N_i$ in the valid bedrooms is between 3 and 12 with 21 object categories, i.e., $K_c = 21$. While for living and dining rooms, $N_i$ varies from 3 to 21 and $K_c = 24$. We use the same data split for training and evaluation as ATISS (Paschalidou et al., 2021).

The original 3D-FRONT dataset does not contain any descriptions of room layout or object appearance details. In order to advance research in the field of text-conditional indoor scene generation, we carefully curate a high-quality dataset with paired scenes and instructions for interior design through a multi-step process:

1. **Spatial Relation Extraction**: View-dependent spatial relations are initially extracted from the 3D-FRONT dataset using predefined rules similar to Johnson et al. (2018) and Luo et al. (2020), which are listed in Appendix A.1.

2. **Object Captioning**: We further enhance the dataset by providing captions to objects using BLIP (Li et al., 2022), a powerful model pretrained for vision-language understanding, given furniture 2D thumbnail images from the original dataset (Fu et al., 2021b).

3. **Caption Refinement**: As generated captions may not always be accurate, we filter them with corresponding ground-truth categories using ChatGPT (Ouyang et al., 2022; OpenAI, 2023), a large language model fine-tuned for instruction-based tasks. This results in accurate and expressive descriptions of each object in the scene. The prompt and hyperparameters for ChatGPT to filter captions are provided in Appendix A.2.

4. **Instruction Generation**: The final instructions for scene synthesis are derived from $1 \sim 2$ randomly selected "(subject, relation, object)" triplets obtained during the first extraction process. Verbs and conjunctions within sentences are also randomly picked to maintain diversity and fluency.

To facilitate future research and replication, the processing scripts and the processed dataset can be found in `https://chenguolin.github.io/projects/InstructScene`.

## A.1    RELATION DEFINATIONS

We define 11 relationships in a 3D space as listed in Table 4. Assume $X$ and $Y$ span the ground plane, and $Z$ is the vertical axis. We use `Center` to represent the coordinates of a 3D bounding box's center. `Height` is the $Z$-axis size of a bounding box. Relative orientation is computed as $\theta_{so} := \text{atan2}(Y_s - Y_o, X_s - X_o)$, where $s$ and $o$ respectively refer to "subject" and "object" in a relationship. $d(s, o)$ is the ground distance between $s$ and $o$. Inside$(s, o)$ indicates whether the subject center is inside the ground bounding box of the object.

## A.2    CAPTION REFINEMENT BY CHATGPT

The generated object captions from BLIP are refined by ChatGPT (`gpt-3.5-turbo`). Our prompt to ChatGPT is provided in Table 5. We set the hyperparameter `temperature` and `top_p` for text generation to `0.2` and `0.1` respectively, encouraging more deterministic and focused outputs.

# B    IMPLEMENTAION DETAILS

## B.1    BASELINE DETAILS

We choose two prominent methods for comparison: (1) ATISS (Paschalidou et al., 2021)[1], an autoregressive model that sequentially generates unordered object sets; (2) DiffuScene (Tang et al., 2023)[2], a Gaussian diffusion model that treats scene attributes as continuous 2D matrices.

---

[1]`https://github.com/nv-tlabs/ATISS`
[2]`https://github.com/tangjiapeng/DiffuScene`

Table 4: Rules to determine the spatial relationships between objects.

| Relationship | Rule |
|---|---|
| Left of | $(\theta_{so} \geq \frac{3\pi}{4}$ or $\theta_{so} < -\frac{3\pi}{4})$ and $1 < d(s,o) \leq 3$ |
| Right of | $-\frac{\pi}{4} \leq \theta_{so} < \frac{\pi}{4}$ and $1 < d(s,o) \leq 3$ |
| In front of | $\frac{\pi}{4} \leq \theta_{so} < \frac{3\pi}{4}$ and $1 < d(s,o) \leq 3$ |
| Behind | $-\frac{3\pi}{4} \leq \theta_{so} < -\frac{\pi}{4}$ and $1 < d(s,o) \leq 3$ |
| Closely left of | $(\theta_{so} \geq \frac{3\pi}{4}$ or $\theta_{so} < -\frac{3\pi}{4})$ and $d(s,o) \leq 1$ |
| Closely right of | $-\frac{\pi}{4} \leq \theta_{so} < \frac{\pi}{4}$ and $d(s,o) \leq 1$ |
| Closely in front of | $\frac{\pi}{4} \leq \theta_{so} < \frac{3\pi}{4}$ and $d(s,o) \leq 1$ |
| Closely bebind | $-\frac{3\pi}{4} \leq \theta_{so} < -\frac{\pi}{4}$ and $d(s,o) \leq 1$ |
| Above | $(\mathtt{Center}_{Z_s} - \mathtt{Center}_{Z_o}) > (\mathtt{Height}_s + \mathtt{Height}_o)/2$ and $(\mathrm{Inside}(s,o)$ or $\mathrm{Inside}(o,s))$ |
| Below | $(\mathtt{Center}_{Z_o} - \mathtt{Center}_{Z_s}) > (\mathtt{Height}_s + \mathtt{Height}_o)/2$ and $(\mathrm{Inside}(s,o)$ or $\mathrm{Inside}(o,s))$ |
| None | $d(s,o) > 3$ |

Table 5: Prompt for ChatGPT to refine raw object descriptions.

Given a description of furniture from a captioning model and its ground-truth category, please combine their information and generate a new short description in one line. The provided category must be the descriptive subject of the new description. The new description should be as short and concise as possible, encoded in ASCII. Do not describe the background and counting numbers. Do not describe size like 'small', 'large', etc. Do not include descriptions like 'a 3D model', 'a 3D image', 'a 3D printed', etc. Descriptions such as color, shape and material are very important, you should include them. If the old description is already good enough, you can just copy it. If the old description is meaningless, you can just only include the category. For example: Given 'a 3D image of a brown sofa with four wooden legs' and 'multi-seat sofa', you should return: a brown multi-seat sofa with wooden legs. Given 'a pendant lamp with six hanging balls on the white background' and 'pendant lamp', you should return: a pendant lamp with hanging balls. Given 'a black and brown chair with a floral pattern' and 'armchair', you should return: a black and brown floral armchair. The above examples indicate that you should delete the redundant words in the old description, such as '3D image', 'four', 'six' and 'white background', and you must include the category name as the subject in the new description. The old descriptions is '{BLIP caption}', its category is '{ground-truth category}', the new descriptions should be:

We re-implement and augment these methods based on their official GitHub repositories to support instruction-driven scene synthesis and quantized semantic feature generation. In the case of ATISS, we replace the [START] token, which originally is the room mask feature, with a learnable token, as we condition scene synthesis on instruction prompts rather than room masks. The augmented ATISS predicts quantized feature indices after class label sampling, and they are subsequently utilized to predict the remaining scene attributes. Instead, quantized features are one-hot encoded in DiffuScene, allowing them to be diffused and denoised in a continuous space alongside other attributes.

To maintain a fair comparison, we use the same experimental settings across all methods, including network architectures, training hyperparameters, object retrieval procedures, rendering schemes, etc.

### B.2 MODEL DETAILS

We use 5-layer and 8-head Transformers with 512 attention dimensions and a dropout rate of 0.1 for all generative models in this work. They are trained by the AdamW optimizer (Loshchilov & Hutter, 2018) for 500,000 iterations with a batch size of 128, a learning rate of 1e-4, and a weight decay of 0.02. Exponentially moving average (EMA) technique (Polyak & Juditsky, 1992; Ho et al., 2020) with a decay factor of 0.9999 is utilized in the model parameters.

We adopt OpenShape `pointbert-vitg14-rgb` (Liu et al., 2023b)[3] to extract 3D object semantic features $\mathbf{f} \in \mathbb{R}^{1280}$. It is a recently introduced 3D RGB point cloud encoder that aligns with the pretrained CLIP `ViT-bigG/14` multi-modal features (Cherti et al., 2023), enabling the simultaneous representation of visual appearances and geometric shapes. The codebook $\mathcal{Z}$ has a size of $64$ and a dimension of $512$. We use $4$ ordered indices to quantize OpenShape features. $f$VQ-VAE is trained on over $4,000$ 3D objects found in the filtered 3D-FRONT scenes (Fu et al., 2021a;b). We use the frozen text encoder in CLIP `ViT-B/32` (Radford et al., 2021)[4] to extract instruction features for all experiments. Regarding the loss weights $\lambda_f$ and $\lambda_e$ in Equation 6, we do not tune and simply fix them as $1$ and $10$ respectively to ensure that the three terms in the loss are of comparable numerical magnitudes.

Code for both training and evaluation can be found in `https://chenguolin.github.io/projects/InstructScene`.

### B.3 EVALUATION DETAILS

**Blender Rendering**  After retrieving objects from a 3D database (Fu et al., 2021b), we use Blender (Community, 2018) with the `CYCLES` engine to render high-quality images for 3D scenes. Our rendering script is adapted from the one available at `https://github.com/allenai/objaverse-rendering/blob/main/scripts/blender_script.py`. The images for evaluation are rendered from a top-down perspective in $256 \times 256$ resolutions. We maintain a camera distance of `1.2`, a filter width of `0.1`, and use the `RGB` color mode. Other hyperparameters are set in accordance with the referenced script. Sizes of floor plans are adapted across scenes to include all objects, and their textures are fixed to ensure the choice does not introduce any bias in evaluations.

**Computation of Metrics**  FID, FID$^{\text{CLIP}}$ and KID scores are computed by the `clean-fid` library (Parmar et al., 2022)[5]. Lower scores derived from these metrics indicate a higher degree of similarity between the learned distributions and real ones. Following Paschalidou et al. (2021), we fine-tuned an AlexNet (Krizhevsky et al., 2012) that had been pretrained on ImageNet to classify the rendered images of synthesized scenes as well as those of ground-truth scenes. The scene classification accuracy (SCA) that approaches 50% signifies better generation performance.

## C  ADDITIONAL RESULTS

### C.1 INSTRUCTION-DRIVEN SCENE SYNTHESIS

We present visualizations of instruction-driven synthesized bedrooms, living rooms, and dining rooms in Figure 5, 6 and 7. Besides the quantitative evaluations shown in Table 1, these qualitative visualizations also evident the superiority of our method over previous state-of-the-art approaches in terms of adherence to instructions and generative quality.

### C.2 ZERO-SHOT APPLICATIONS

We present visualizations illustrating various zero-shot instruction-driven applications, including stylization, re-arrangement, completion, and unconditional 3D scene synthesis in Figure 8, 9, 10 and 11 respectively. We find that the autoregressive model ATISS tends to generate redundant objects, resulting in chaotic synthesized scenes. DiffuScene encounters challenges in accurately modeling object semantic features, often yielding objects that lack coherence in terms of style or pairing, thereby diminishing the aesthetic appeal of the synthesized scenes. Moreover, both of these baseline models frequently struggle to follow the provided instructions during conditional generation. In contrast, our approach demonstrates a notable capability to generate highly realistic 3D scenes that concurrently adhere to the provided instructions.

---

[3]`https://github.com/Colin97/OpenShape_code`
[4]`https://github.com/openai/clip`
[5]`https://github.com/GaParmar/clean-fid`

## C.3 FEATURE RECOVERY

We conduct two additional experiments to further validate our method: (1) masking the semantic feature of one object and utilizing a pretrained semantic graph prior for recovery: $p_\phi(\mathbf{f}_i|\mathbf{f}_{/i}, c, \mathbf{t}, \mathbf{s}, r)$; (2) masking semantic features of all objects except one and again using the pretrained semantic graph prior for recovery: $p_\phi(\mathbf{f}_{/i}|\mathbf{f}_i, c, \mathbf{t}, \mathbf{s}, r)$. $\mathbf{f}_{/i}$ means semantic features of all objects except the $i$-th one. Instructions for both experiments are set to none. Visualization results are presented in Figure 12 and 13 respectively.

These results indicate the diversity of our method and highlight that semantic graph prior could effectively capture stylistic information and object co-occurrences from the training data. Our method trends to generate style consistent and thematic harmonious scenes, e.g., chairs and nightstands in a suit, and matched color palettes and cohesive artistic style.

## C.4 DIVERSITY

We provide examples of a diverse set of scenes generated from a single prompt and the same semantic graph in Figure 14 and 15 respectively, showcasing the diversity of our generative method.

## C.5 INSTRUCTSCENE WITHOUT SEMANTIC FEATURES

We observed a significant decline in the appearance controllability and style consistency of generated scenes when semantic features were omitted. We include these degraded visualization results in Figure 16 and 17.

It arises from the fact that, without semantic features, the generative models solely focus on modeling the distributions of layout attributes, i.e., categories, translations, rotations, and scales. This exclusion of semantic features results in generated objects whose occurrences and combinations lack awareness of object style and appearance, which are crucial elements in scene design.

## C.6 RUNTIME COMPARISON

In the default settings ($T = 100 + 10$), our method takes about 12 seconds to generate a batch of 128 living rooms by our method on a single A40 GPU. In comparison, ATISS (Paschalidou et al., 2021) takes 3 seconds, and DiffuScene (Tang et al., 2023) requires 22 seconds.

It's noteworthy that our method can be significantly accelerated by reducing the number of diffusion time steps. For instance, setting $T = 20 + 5$ reduces the runtime to 3 seconds without a noticeable decline in performance. The impact of diffusion time steps is investigated in Sec. 5.5.1. We believe with more advanced diffusion techniques, diffusion models can be more effective and efficient than autoregressive models, especially for complex scenes.

## D LIMITATIONS AND FUTURE WORK

Although our method significantly enhances the controllability and fidelity of 3D indoor scene synthesis, it still has some limitations. First, despite our best efforts to ensure the accuracy of the proposed instruct-scene pair dataset, 3D-FRONT contains problematic object arrangements and misclassifications even after filtering, as discussed in previous works (Paschalidou et al., 2021; Tang et al., 2023). Our learned prior may consequently inherit these erroneous cases. Meanwhile, the scale of the current 3D scene dataset remains small, with only hundreds of scenes, in contrast to 3D object datasets containing thousands or even millions of samples (Chang et al., 2015b; Deitke et al., 2023). A promising avenue for future research is to expand the scale of the 3D scene dataset or leverage large-scale and well-annotated datasets for 3D objects to establish a new benchmark for 3D scene synthesis. In this work, we only focus on indoor scene synthesis. However, the proposed semantic graph prior, which encapsulates high-level object interactions within a scene, also offers the potential for modeling more intricate outdoor scenes. Furthermore, achieving a fully generative synthesis pipeline is feasible by substituting the object retrieval step with 3D object generative models conditioned on categories and semantic features provided by our graph prior. Lastly, in light of the rapid development of large language models (LLMs), the integration of an LLM into our instruction-driven pipeline holds significant promise for further enhancing generation controllability.

# E    DISCUSSION ON DATASET

While the curated instructions in our proposed dataset are derived from predefined rules, we believe that our model exhibits generalizability to a broader range of instructions. For example, in the stylization task, we utilize instructions in different sentence patterns with training, such as "Let the room be wooden style" and "Make objects in the room black", as illustrated in Figure 8. We also experiment with instructions containing vague location words, like "Put a chair next to a double bed", wherein our method generates corresponding objects in all possible spatial relations (e.g., "left", "right", "front", and "behind").

Nevertheless, INSTRUCTSCENE still faces limitations in comprehending complex text instructions and abstract concepts that do not occur in the curated instructions. For instance, (1) handling instructions with more required triplets, like 4 or 5, poses a challenge. (2) Additionally, identifying the same object within one instruction, such as "Put a table left to a sofa. Then add a chair to the table mentioned before" is also a difficult task. (3) Furthermore, it struggles with abstract concepts such as artistic style, occupants, and functionalities that do not occur in the curated instructions. These limitations are attributed to the CLIP text encoder, which is contrastively trained with image features and tends to capture global semantic information in sentences. Given the rapid development of large language models, we believe the integration of LLMs into the proposed pipeline is a promising research topic.

A viable approach to improve the quality of current instructions involves employing LLMs to refine entire sentences in the proposed dataset or using crowdsourcing to make the dataset curation pipeline semi-supervised. We hope the proposed dataset and creation pipeline could serve as a good starting point for creating high-quality instruction datasets.

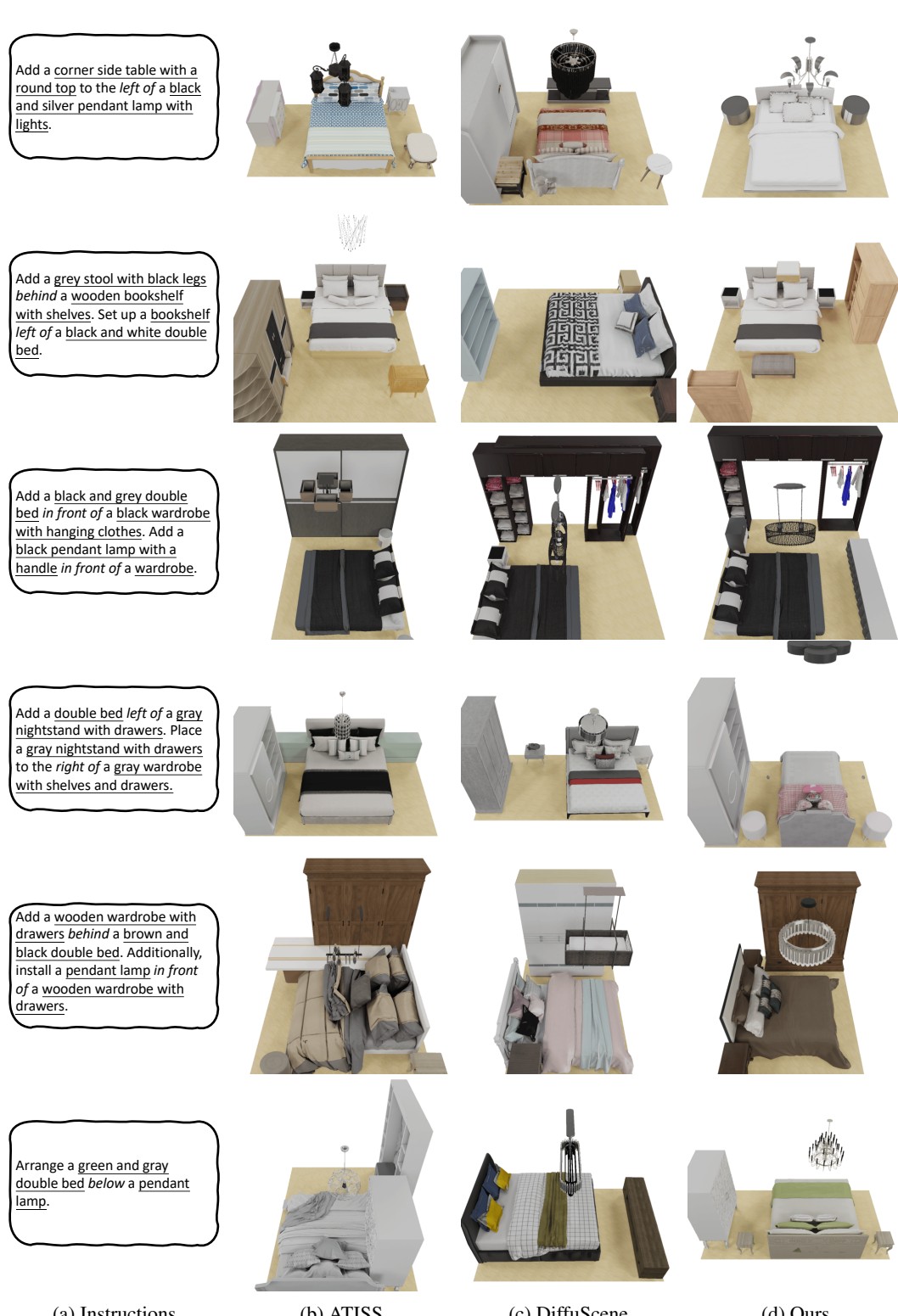

Figure 5: Visualizations for instruction-drive synthesized 3D bedrooms by ATISS (Paschalidou et al., 2021), DiffuScene (Tang et al., 2023) and our method.

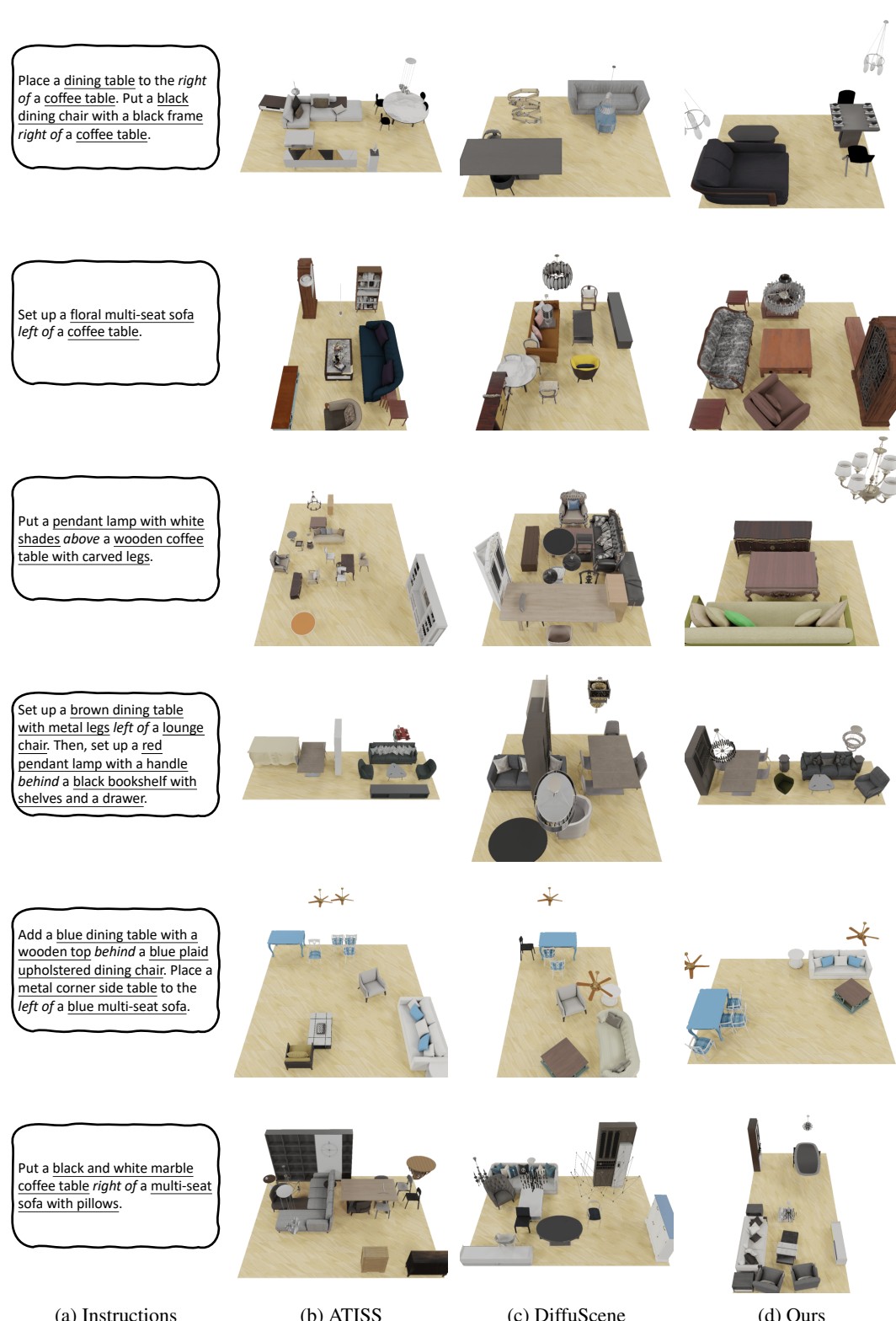

Figure 6: Visualizations for instruction-drive synthesized 3D living rooms by ATISS (Paschalidou et al., 2021), DiffuScene (Tang et al., 2023) and our method.

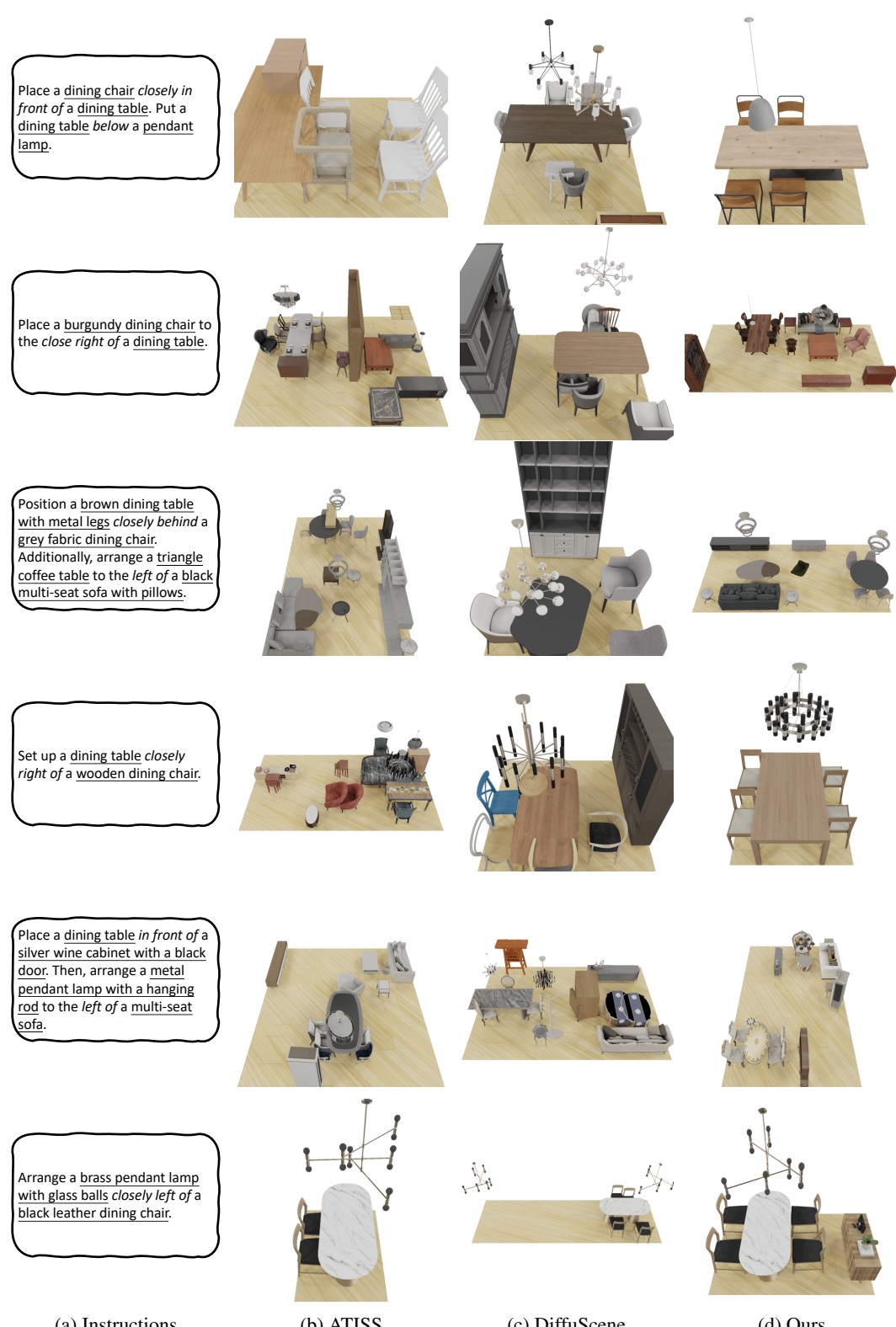

(a) Instructions      (b) ATISS      (c) DiffuScene      (d) Ours

Figure 7: Visualizations for instruction-drive synthesized 3D dining rooms by ATISS (Paschalidou et al., 2021), DiffuScene (Tang et al., 2023) and our method.

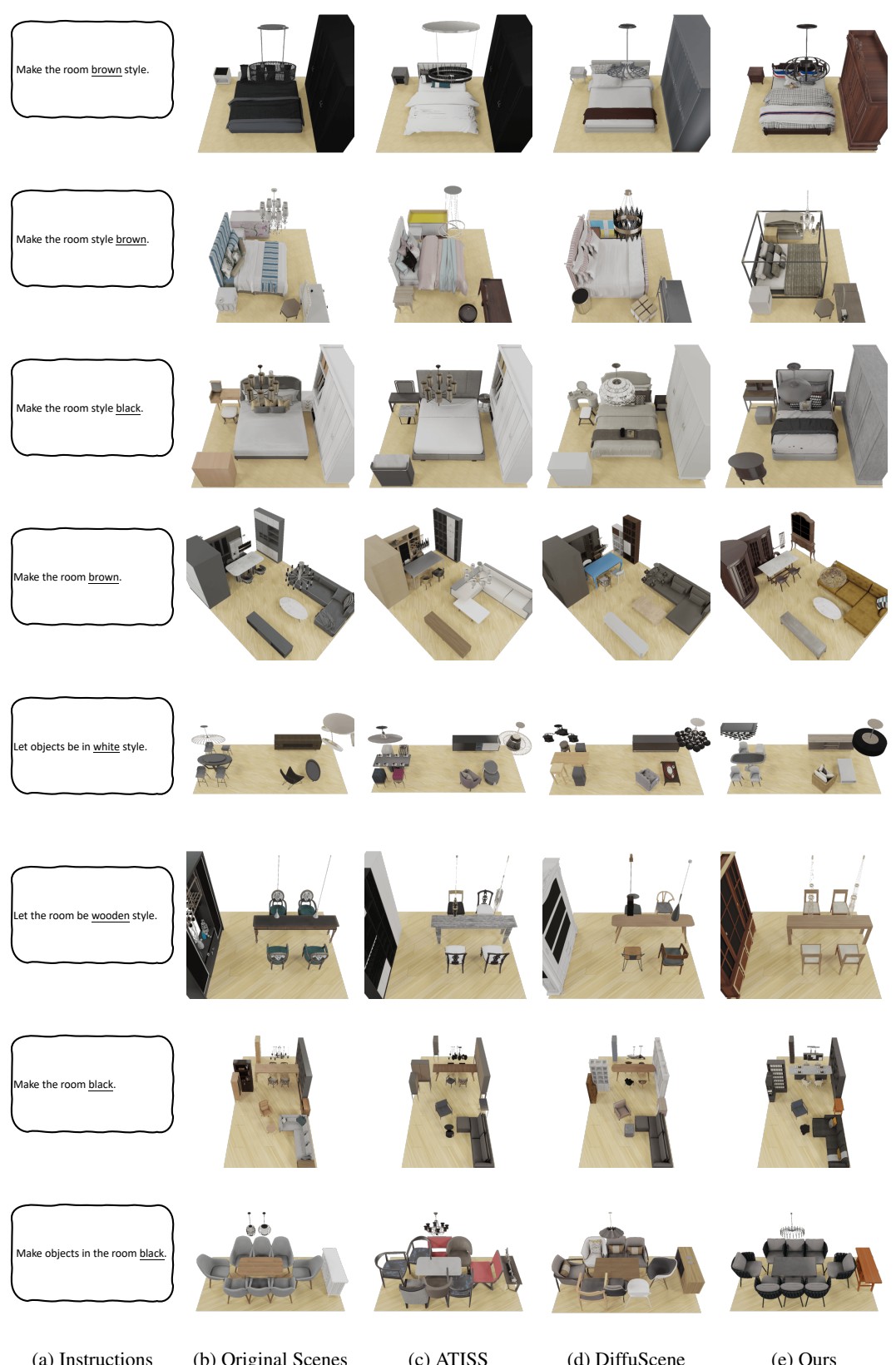

Figure 8: Visualizations for instruction-drive 3D scenes stylization by ATISS (Paschalidou et al., 2021), DiffuScene (Tang et al., 2023) and our method.

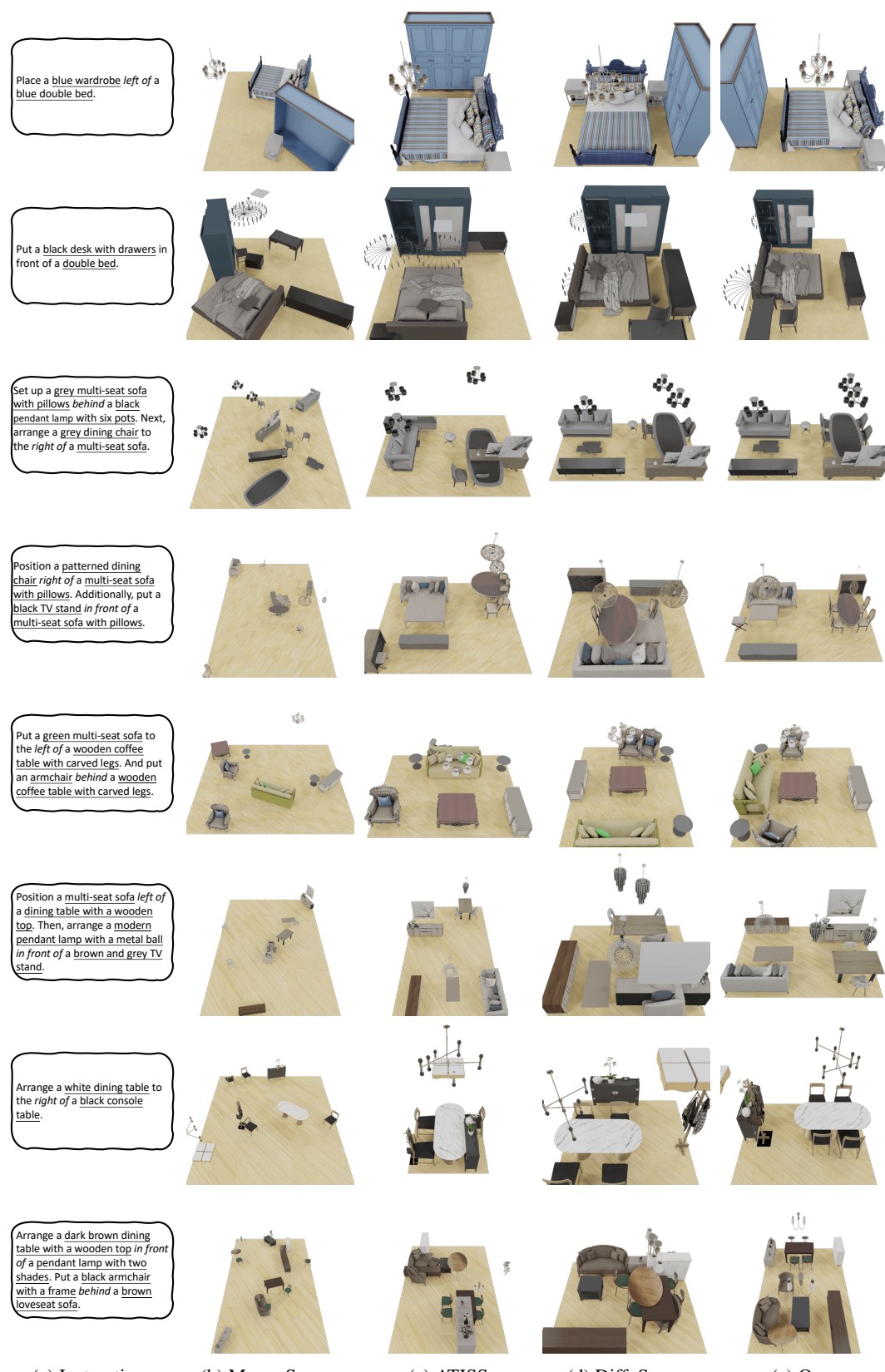

Figure 9: Visualizations for instruction-drive 3D scenes re-arrangement by ATISS (Paschalidou et al., 2021), DiffuScene (Tang et al., 2023) and our method.

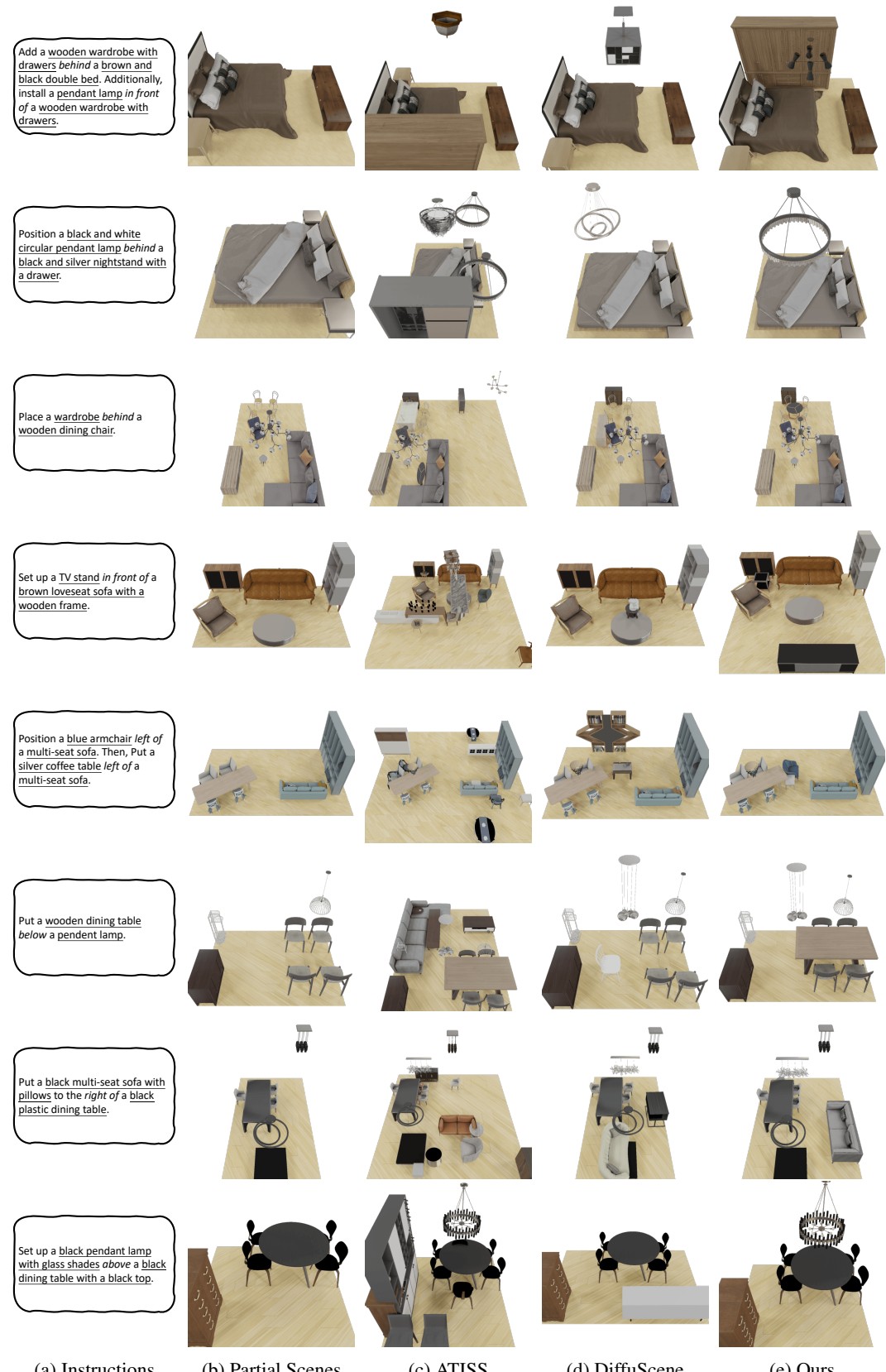

Figure 10: Visualizations for instruction-drive 3D scenes completion by ATISS (Paschalidou et al., 2021), DiffuScene (Tang et al., 2023) and our method.

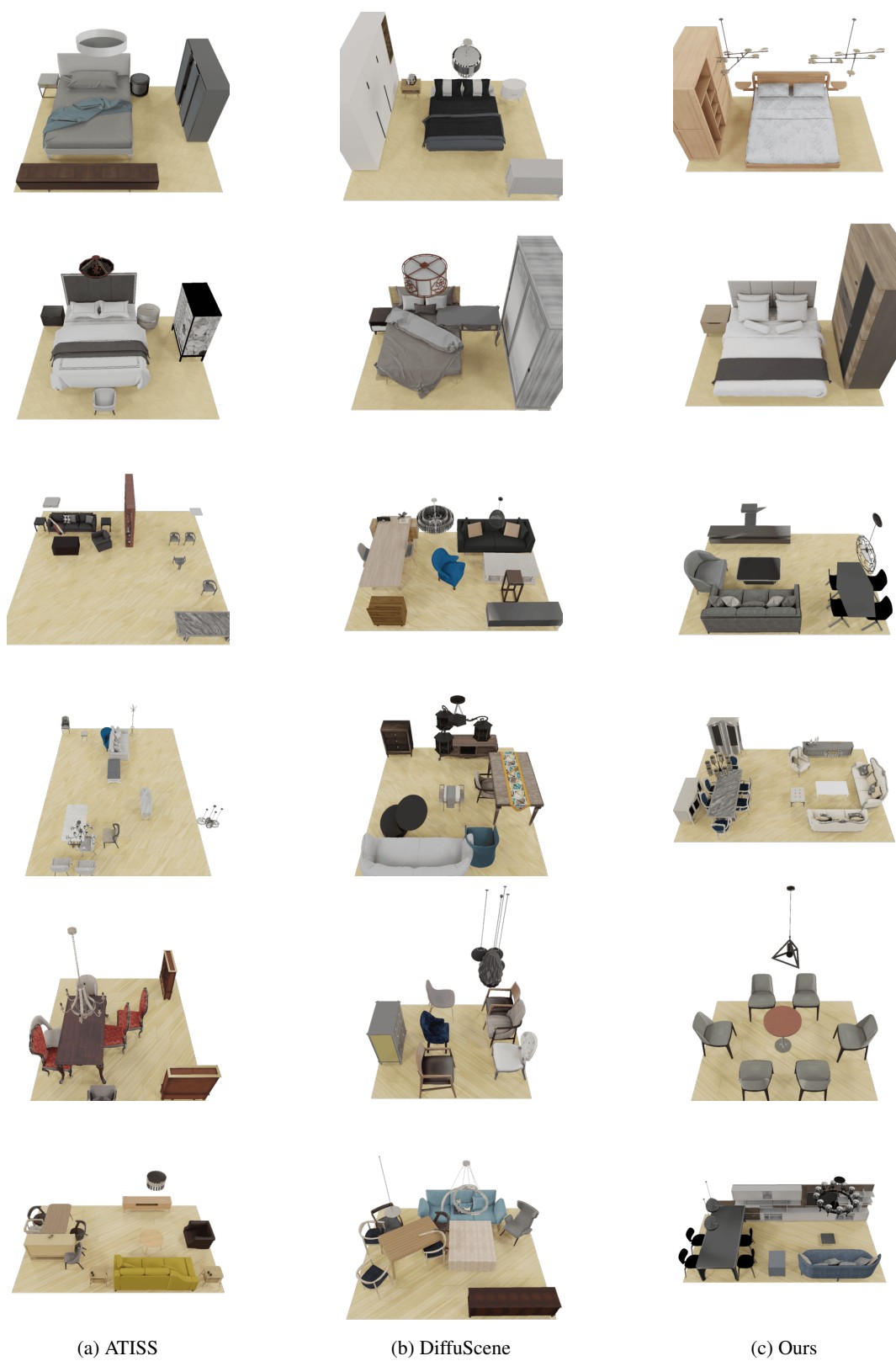

(a) ATISS        (b) DiffuScene        (c) Ours

Figure 11: Visualizations for unconditional 3D scenes stylization by ATISS (Paschalidou et al., 2021), DiffuScene (Tang et al., 2023) and our method.

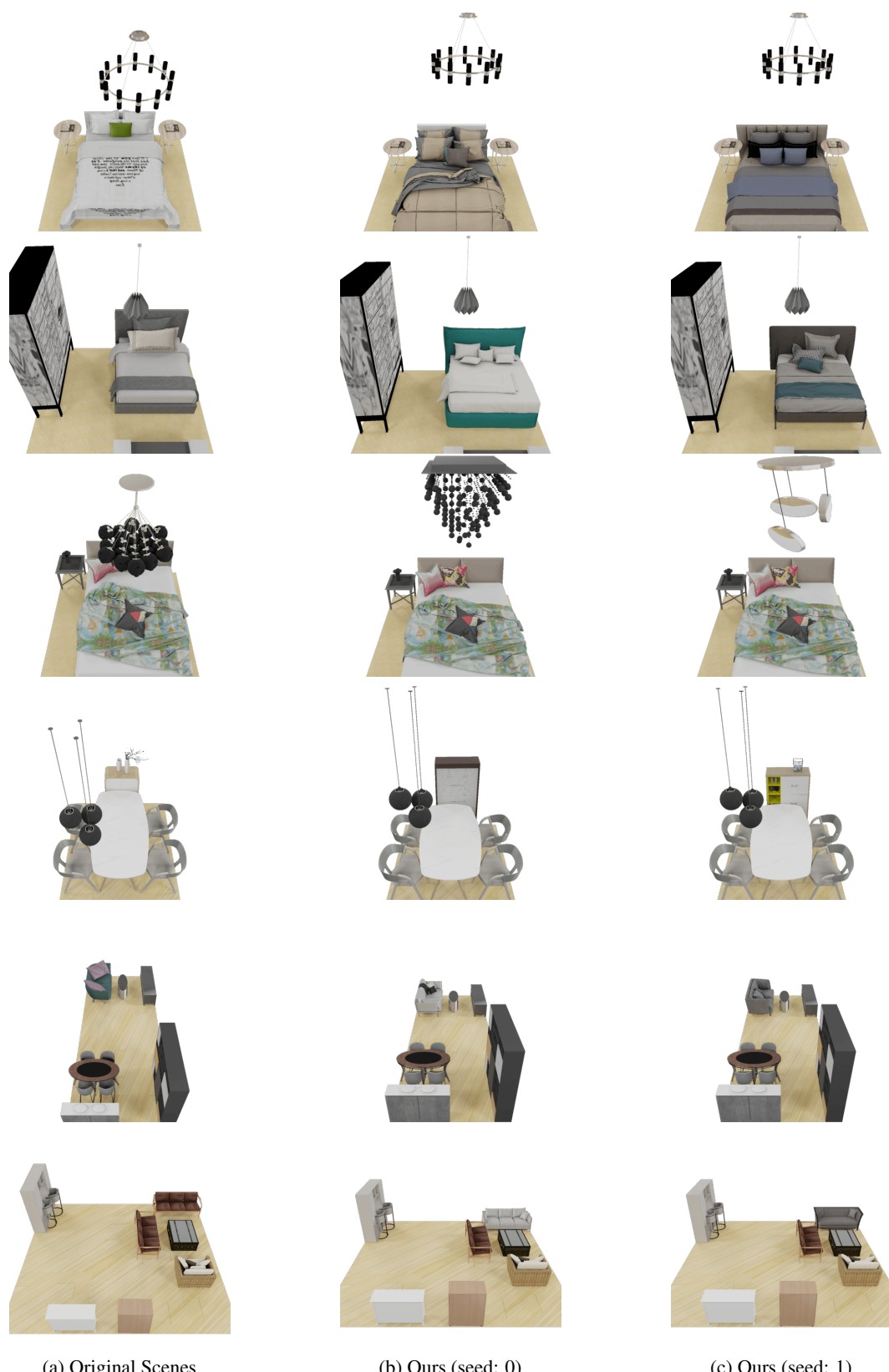

(a) Original Scenes          (b) Ours (seed: 0)          (c) Ours (seed: 1)

Figure 12: Generating the semantic feature of one object without instructions $p_\phi(\mathbf{f}_i|\mathbf{f}_{/i}, c, \mathbf{t}, \mathbf{s}, r)$.

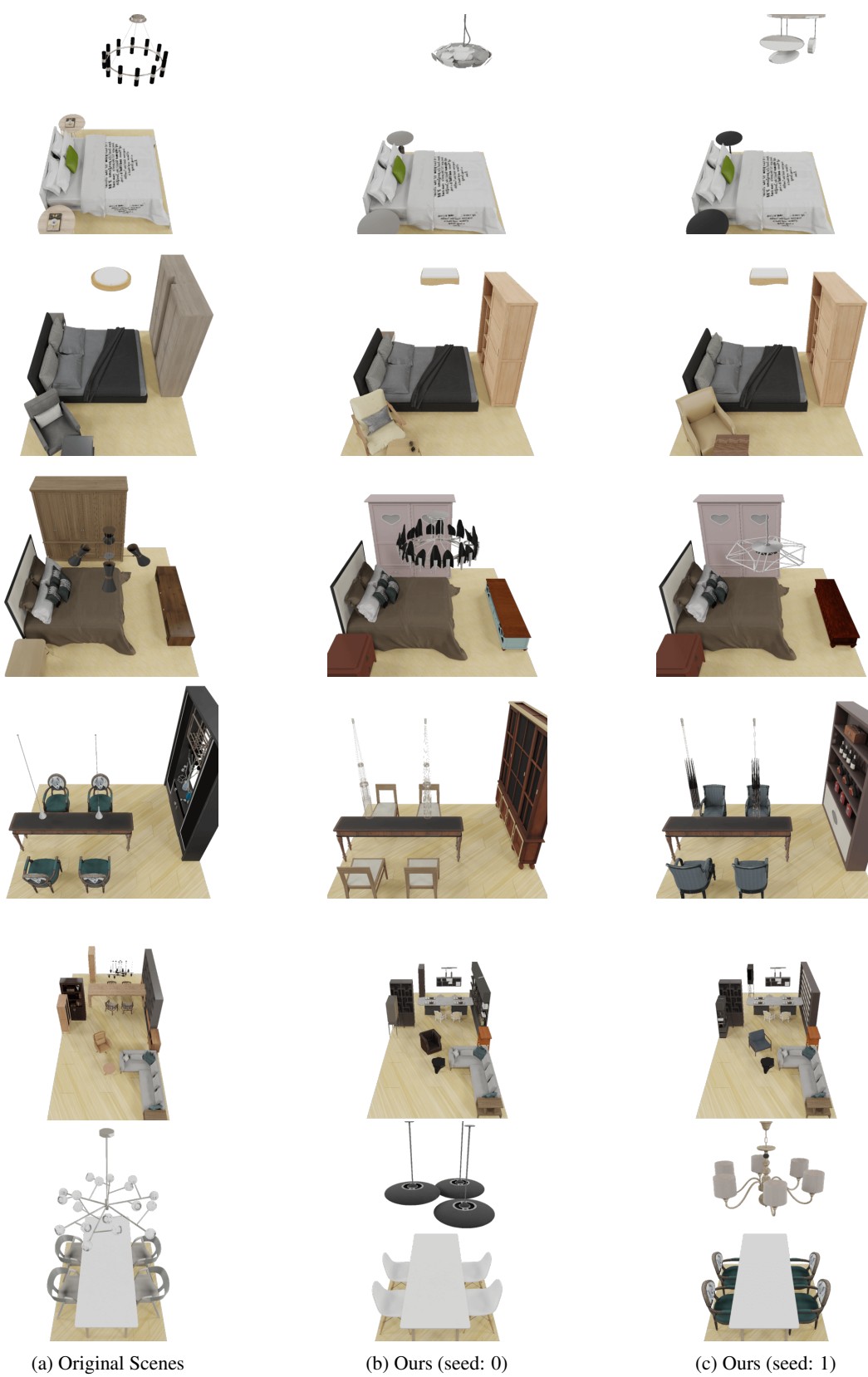

(a) Original Scenes       (b) Ours (seed: 0)       (c) Ours (seed: 1)

Figure 13: Generating semantic features of all objects in a scene except one without instructions $p_\phi(\mathbf{f}_{/i}|\mathbf{f}_i, c, \mathbf{t}, \mathbf{s}, r)$.

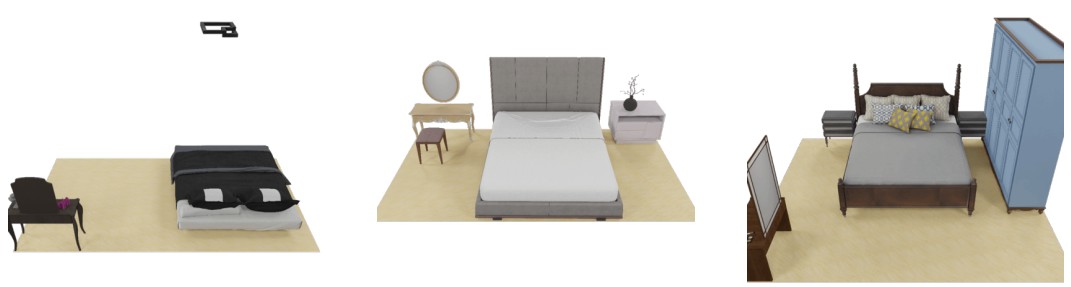

(a) Put a dressing table with a mirror to the left of a double bed.

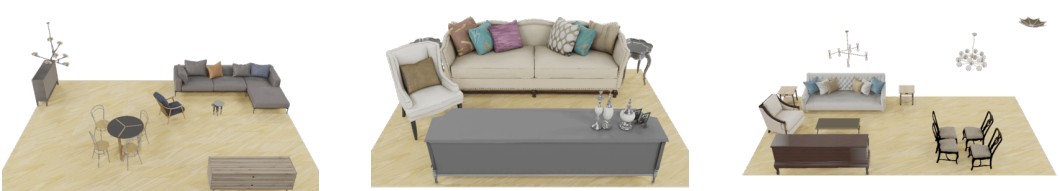

(b) Add a lounge chair in front of a multi-seat sofa.

Figure 14: Examples of a diverse set of scenes generated from a single prompt.

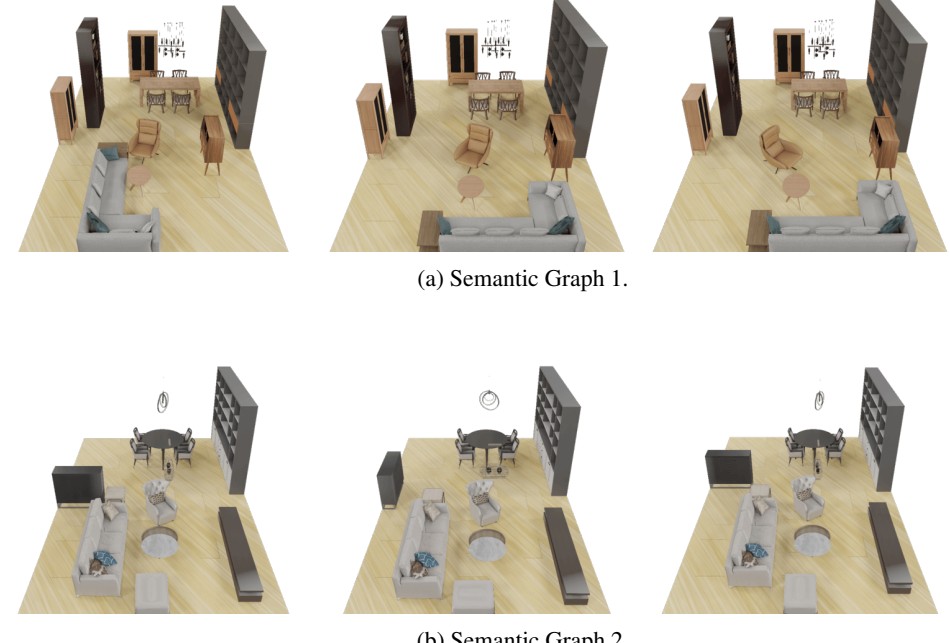

(a) Semantic Graph 1.

(b) Semantic Graph 2.

Figure 15: Examples of a diverse set of scenes generated from the same semantic graph.

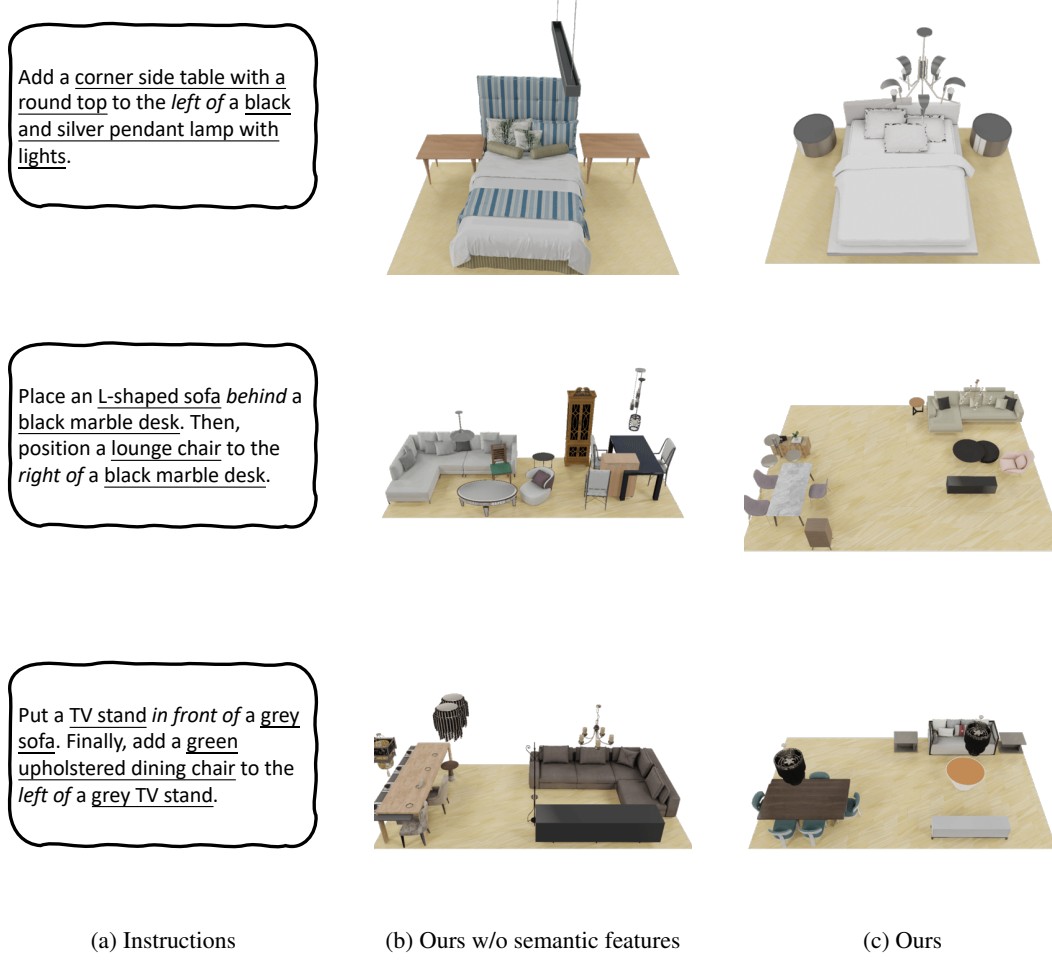

(a) Instructions          (b) Ours w/o semantic features          (c) Ours

Figure 16: Instruction-driven scene synthesis results of INSTRUCTSCENE and its degraded version, which is not encoded with semantic features.

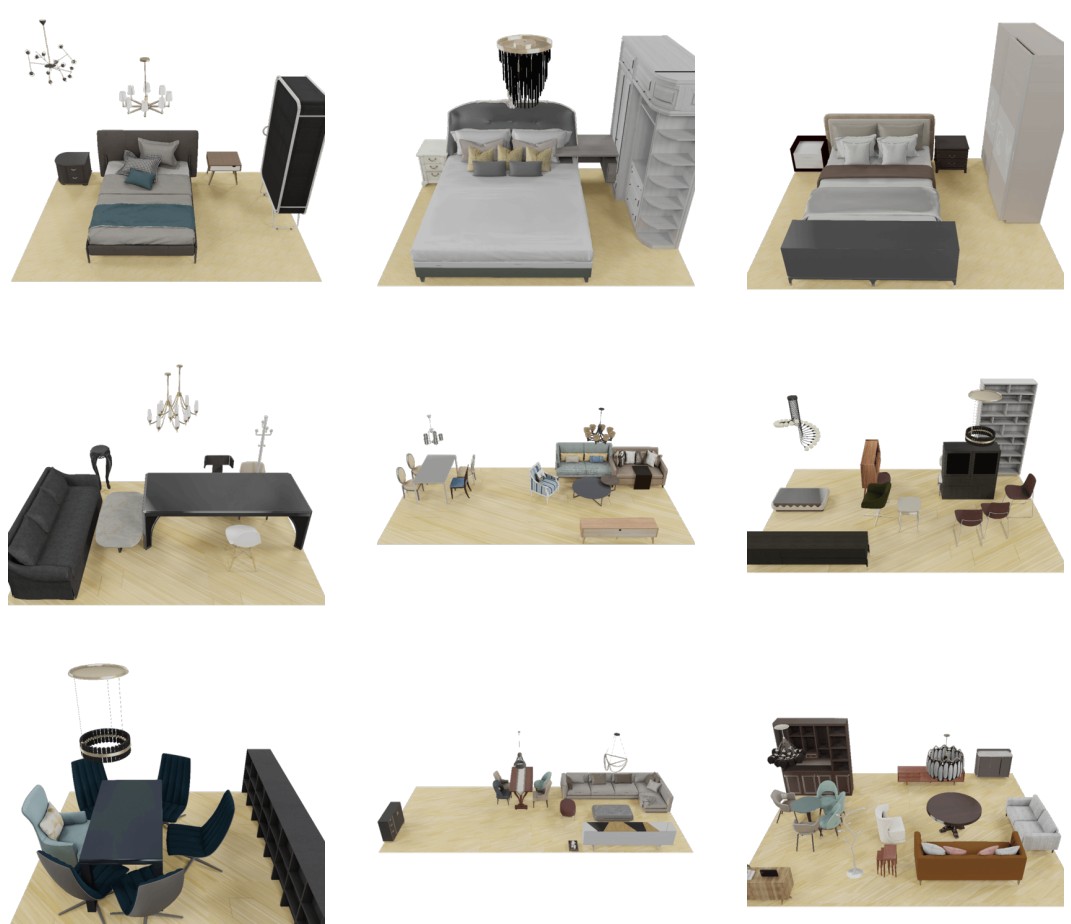

Figure 17: Unconditional scene synthesis results of a degraded version of INSTRUCTSCENE, which is not encoded with semantic features.

