# REBUTTAL SUPPLEMENTARY MATERIAL

# INSTRUCTSCENE: INSTRUCTION-DRIVEN 3D INDOOR SCENE SYNTHESIS WITH SEMANTIC GRAPH PRIOR

We provide more qualitative results in this supplementary material to address concerns raised by some reviewers.

1. Results of generating the semantic feature of one object without instructions $p_\phi(\mathbf{f}_i|\mathbf{f}_{/i}, c, \mathbf{t}, \mathbf{s}, r)$ are presented in Figure 1. $\mathbf{f}_{/i}$ means semantic features of all objects except the $i$-th one. Results of generating semantic features of all objects in a scene except one without instructions $p_\phi(\mathbf{f}_{/i}|\mathbf{f}_i, c, \mathbf{t}, \mathbf{s}, r)$ are presented in Figure 2.

   These results indicate the diversity of our method and highlight that semantic graph prior could effectively capture stylistic information and object co-occurrences from the training data. Our method trends to generate style consistent and thematic harmonious scenes, e.g., chairs and nightstands in a suit, and matched color palettes and cohesive artistic style.

2. Instruction-driven and unconditional scene synthesis results of a degraded version of IN- STRUCTSCENE, which is not encoded with semantic features, are presented in Figure 3 and Figure 4 respectively.

   We observe a significant decline in the appearance controllability and style consistency of gen- erated scenes when semantic features were omitted. It arises from the fact that, without semantic features, the generative models solely focus on modeling the distributions of layout attributes, i.e., categories, translations, rotations, and scales. This exclusion of semantic features results in generated objects whose occurrences and combinations lack awareness of object style and appearance, which are crucial elements in scene design.

3. Examples of a diverse set of scenes generated from a single prompt and the same semantic graph are presented in Figure 5 and Figure 6 respectively, showcasing the diversity of the proposed generative method.

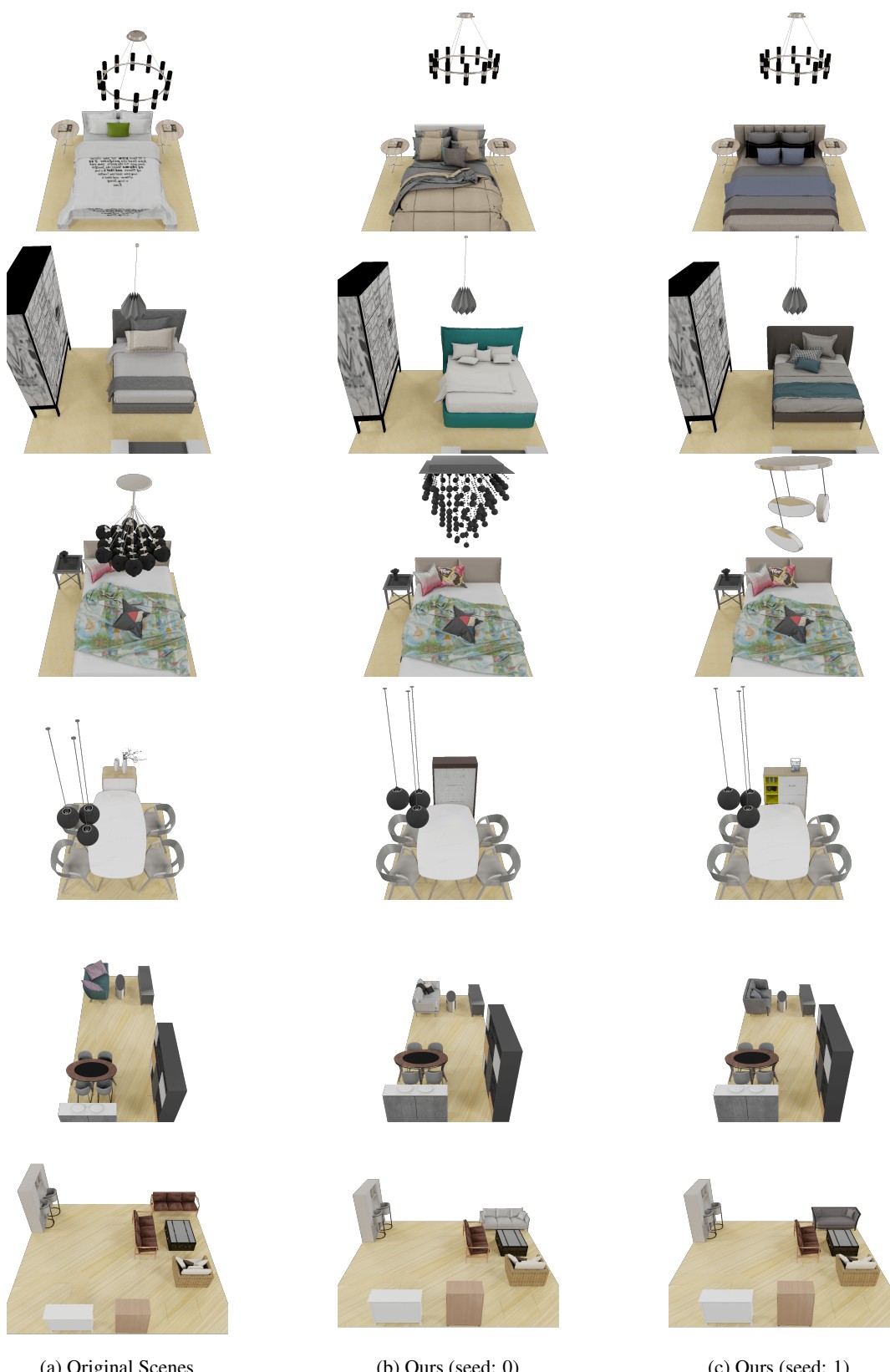

(a) Original Scenes        (b) Ours (seed: 0)        (c) Ours (seed: 1)

Figure 1: Generating the semantic feature of one object without instructions $p_\phi(\mathbf{f}_i | \mathbf{f}_{/i}, c, \mathbf{t}, \mathbf{s}, r)$.

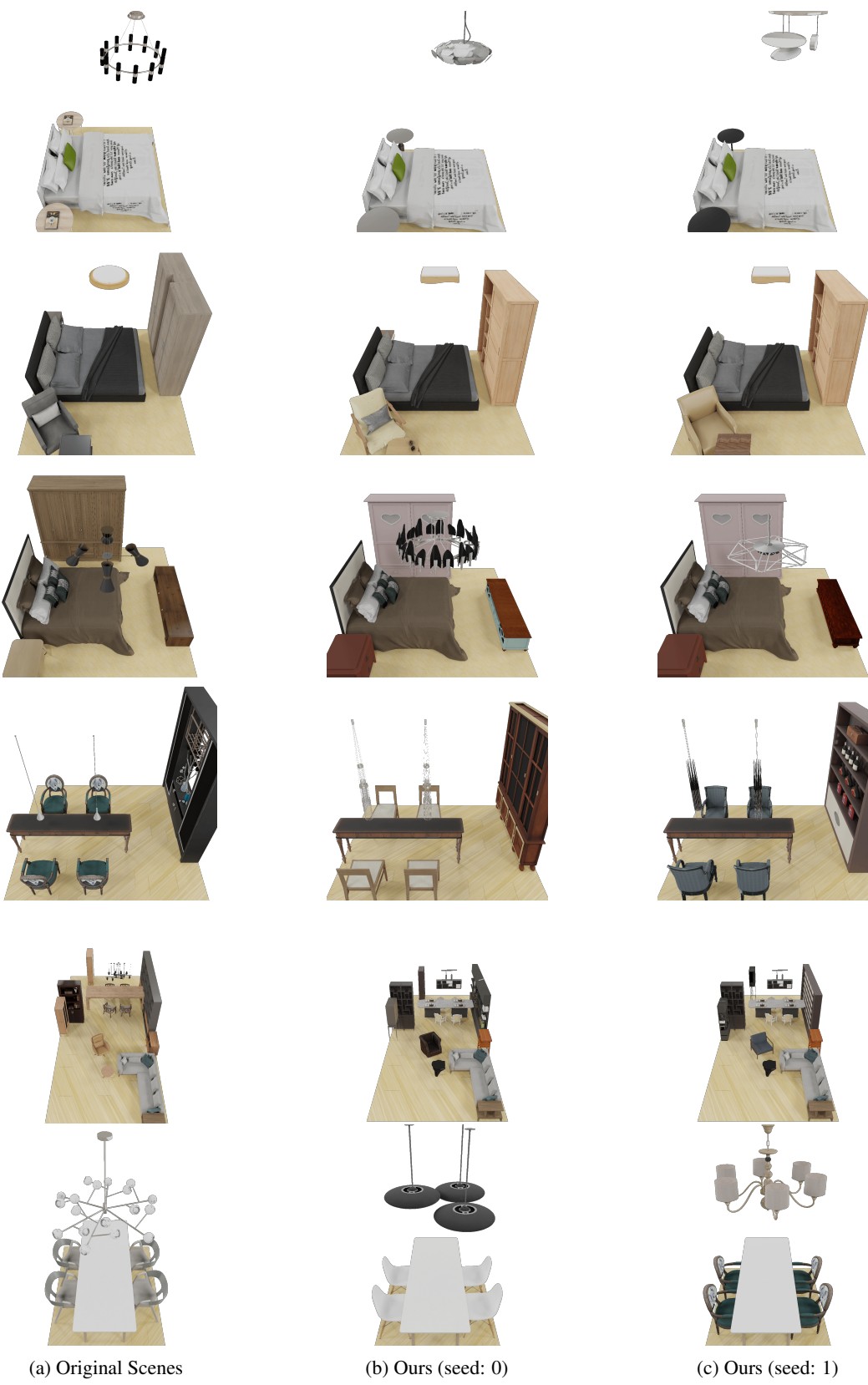

(a) Original Scenes       (b) Ours (seed: 0)       (c) Ours (seed: 1)

Figure 2: Generating semantic features of all objects in a scene except one without instructions $p_\phi(\mathbf{f}_{/i}|\mathbf{f}_i, c, \mathbf{t}, \mathbf{s}, r)$.

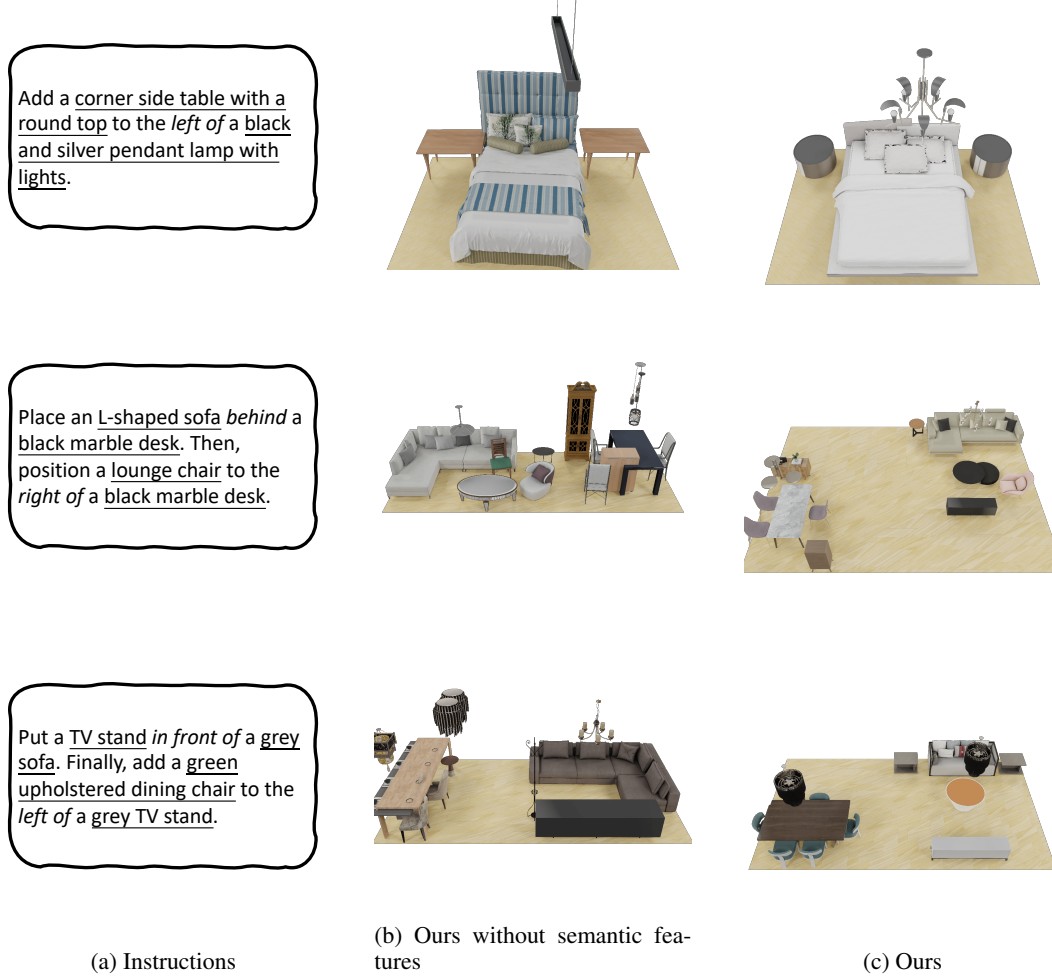

(a) Instructions

(b) Ours without semantic features

(c) Ours

**Add a** corner side table with a round top **to the** *left of* **a** black and silver pendant lamp with lights.

**Place an** L-shaped sofa *behind* **a** black marble desk. **Then, position a** lounge chair **to the** *right of* **a** black marble desk.

**Put a** TV stand *in front of* **a** grey sofa. **Finally, add a** green upholstered dining chair **to the** *left of* **a** grey TV stand.

Figure 3: Instruction-driven scene synthesis results of INSTRUCTSCENE and its degraded version, which is not encoded with semantic features.

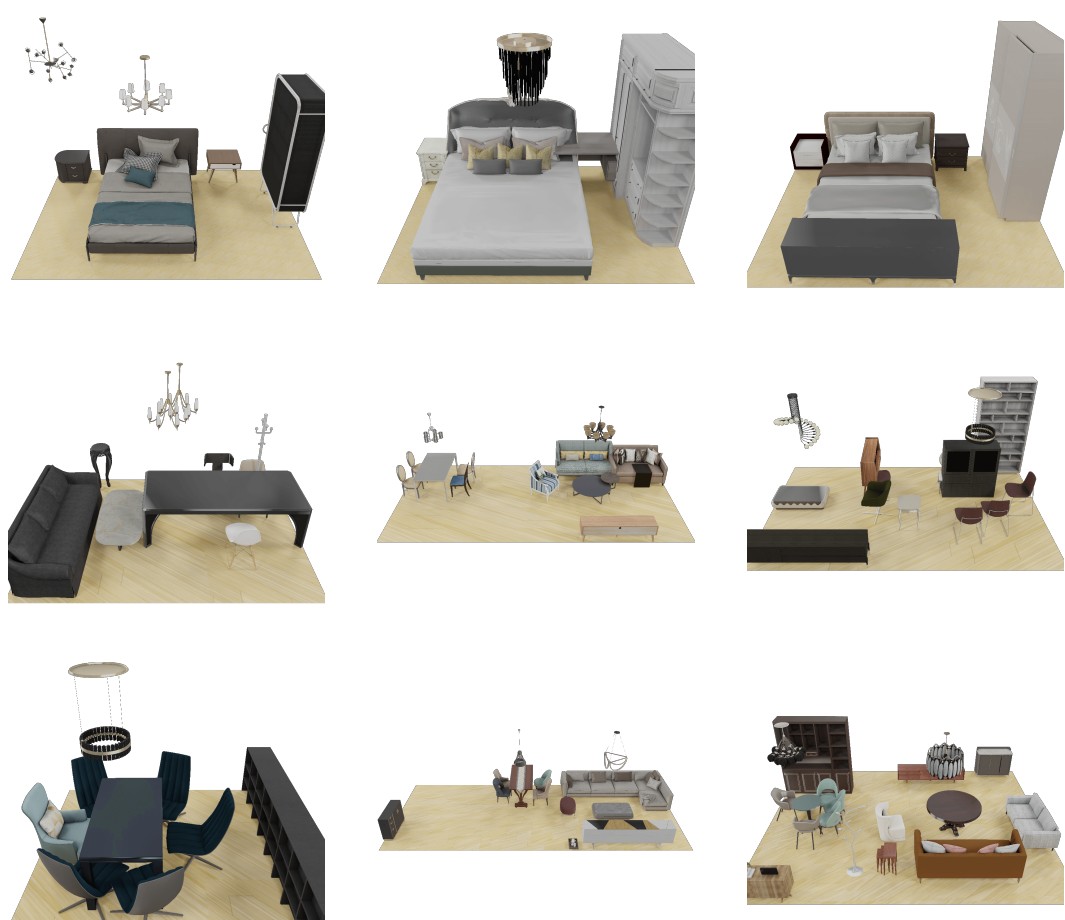

Figure 4: Unconditional scene synthesis results of a degraded version of INSTRUCTSCENE, which is not encoded with semantic features.

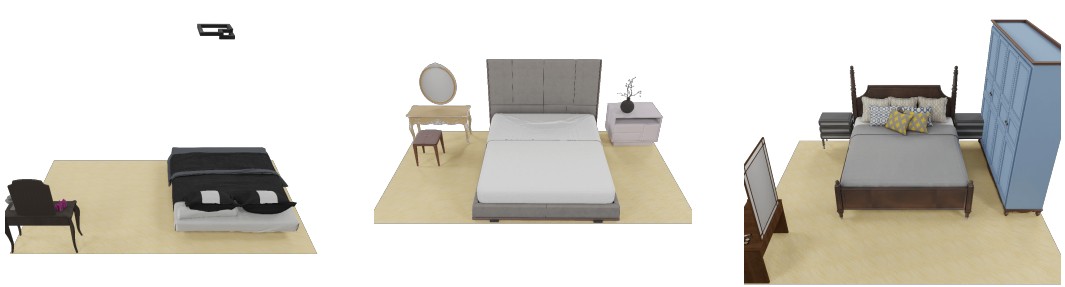

(a) Put a dressing table with a mirror to the left of a double bed.

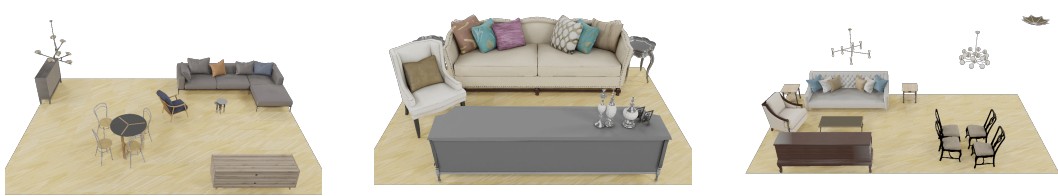

(b) Add a lounge chair in front of a multi-seat sofa.

Figure 5: Examples of a diverse set of scenes generated from a single prompt.

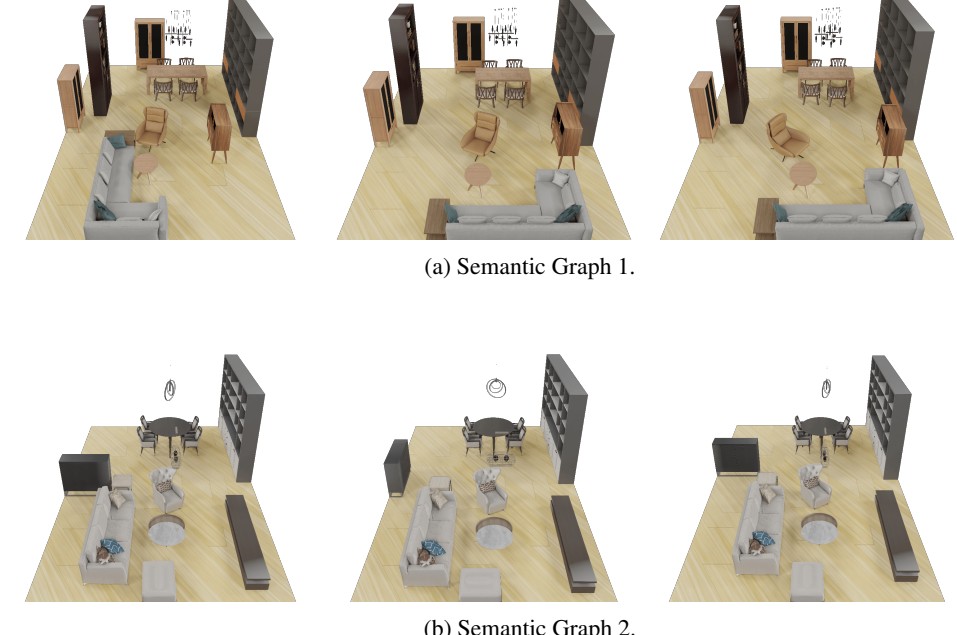

(a) Semantic Graph 1.

(b) Semantic Graph 2.

Figure 6: Examples of a diverse set of scenes generated from the same semantic graph.