# OpenReview forum: "InstructScene: Instruction-Driven 3D Indoor Scene Synthesis with Semantic Graph Prior"
_ICLR.cc/2024/Conference — ICLR 2024 spotlight_

### Official Review · Reviewer_rQD7 · 2023-10-29

**Soundness:** 4 excellent
**Presentation:** 4 excellent
**Contribution:** 3 good
**Rating:** 8
**Confidence:** 4

**Summary:**

The paper proposes InstructScene, which does 3D indoor scene synthesis with semantic graph prior. It first quantizes CLIP features using a transformer with learnable tokens, use a frozen text encoder and graph transformer generate the semantic graph, and finally uses a graph transformer to decoder the final layout. Experiments are done on 3D-Front following the prior works although instructions are re-generated. With the semantic graph prior,  InstructScene outperforms state-of-the-art methods significantly on controllability. It also shows zero-shot generalization to a few other tasks.

**Strengths:**

- The problem is meaningful. Improving controllability critical to 3D scene generation, as we have already seen a lot of 3D scene generation models.
- The method is well-designed.
- Experiments are extensive.
- The paper is very well-written and easy to follow. It provides enough mathematical details, as well as comprehensive qualitative results. I enjoy reading the paper.

**Weaknesses:**

InstructScene is proposed to handle natural language instructions. However, the instruction does not seem to support arbitrary instructions. In qualitative results, the instruction always explicitly mentions one of these relationships.

>  (Section 5.2) we use a metric named “instruction recall” (iRecall), which quantifies the proportion of the required triplets “(subject, relation, object)” occurring in synthesized scenes to all provided in instructions

It's not clear to me how iRecall is computed. To the best of my understanding, iRecall computes the recall, where the prediction is generated scene graph triplets and the ground truth is the instruction triplets. However, how do you get instruction triplets? Let's say the instruction is "Add a corner side table with a round top to the left of a black and silver pendant lamp with lights".  It looks non-trivial to extract triplets. Or do you use relationship keywords to extract?

Additional comments:
- Table 4: bebind -> behind

**Questions:**

- Can you confirm how iRecall is computed?
- Can you comment on what will happen if we give "arbitrary instructions" (11 pre-defined relationships are not available)?

---

> ### Author Response · Authors · 2023-11-15
> **Response to Reviewer rQD7**
>
> Thank you for your insightful and valuable feedback. It is really inspiring to know that you think our work is meaningful and well-designed, our experiments are extensive, and the writing is easy to follow. Below are our clarifications for your concerns. We value your input and would love to hear any follow-up you have to the response.
>
> **1. Can you confirm how iRecall is computed?**
>
> It is right that “iRecall is the proportion of the required triplets in the generated scenes and the ground truth in instruction triplets”.
>
> However, the triplets are not extracted from instructions. Instead, instructions in our dataset are derived from 1~2 randomly selected triplets. Thus, we have the ground-truth triplets of corresponding instructions for iRecall evaluation. More details about the dataset are provided in Appendix A.
>
> **2. Can you comment on what will happen if we give “arbitrary instructions”?**
>
> While the curated instructions in our proposed dataset are derived from predefined rules, we believe that our model exhibits generalizability to a broader range of instructions. We use instructions automatically generated in a certain pattern only for evaluation convenience.
>
> The generalizability owes to the pretrained CLIP text encoder, which is adapted to extract semantically meaningful features of text inputs, instead of using a language parser like the early studies of 3D scene synthesis methods [1, 2] that can only handle sentences in limited expressions. It means the text features that we used from conditioning would be close in the feature space if the input texts have similar meanings.
>
> We utilize instructions in various sentence patterns in the stylization task, such as "Let the room be wooden style" and "Make objects in the room black," as illustrated in Appendix Figure 8. We also try some meaningless “arbitrary instructions”, such as “a cat on the ground” and “a dog wearing a santa hat”, and our method can still generate meaningful and aesthetic 3D scenes like those generated unconditionally shown in Appendix Figure 11. When the inputs are some sentences with vague location words, like “Put a chair next to a double bed.”, our method will generate corresponding objects in all possible spatial relations (e.g., “left”, “right”, “front” and “behind”).
>
> Nevertheless, our model still faces limitations in comprehending complex text instructions. For instance, (1) handling instructions with more required triplets, like 4 or 5, poses a challenge. (2) Additionally, identifying the same object within one instruction, such as "Put a table left to a sofa. Then add a chair to the table mentioned before" is also a difficult task. (3) Furthermore, InstructScene struggles with abstract concepts such as artistic style, occupants, and functionalities that do not occur in the curated instructions. Given the rapid development of large language models (LLMs), we believe the integration of LLMs into the proposed pipeline is a promising research topic.
>
> A viable approach to improve the quality of current instructions involves employing LLMs to refine entire sentences in the proposed dataset or using crowdsourcing to make the dataset curation pipeline semi-supervised. We hope the proposed dataset and creation pipeline could serve as a good starting point for creating high-quality instruction datasets.
>
> We will add this discussion to the revision of this work.
>
>
> ---
> [1] Learning Spatial Knowledge for Text to 3D Scene Generation. EMNLP 2014.
>
> [2] Text to 3D Scene Generation with Rich Lexical Grounding. ACL 2015.

---

> > ### Comment · Reviewer_rQD7 · 2023-11-22
> >
> > Thanks for your response, which confirms my understanding of iRecall is correct, and gives insights for arbitrary text inputs.
> >
> > My concerns have been addressed, and I recommend accepting the paper.

---

### Official Review · Reviewer_qrjz · 2023-10-31

**Soundness:** 3 good
**Presentation:** 1 poor
**Contribution:** 2 fair
**Rating:** 6
**Confidence:** 2

**Summary:**

The paper proposes a semantic graph prior plus a layout decoder system to generate indoor scenes based on natural language instructions. Instruction-scene pairs are generated as a new dataset for the task.

**Strengths:**

- The paper collects a dataset that supplements existing indoor scene datasets with natural language instructions.
- The paper proposes a two-stage method that is shown to be more effective than previous methods in quantitative and qualitative evaluations.
- The proposed method demonstrates some abilities in zero-shot indoor scene editing.

**Weaknesses:**

- I am concerned about the novelty of the proposed method. The semantic graph prior is a combination of existing VQ-VAE and discrete diffusion models. The layout decoder is a plain diffusion model.
- Some claims or arguments about using VQ-VAE are not convincing and require further substantiation. The authors claim that VQ-VAE is used because objects share common characteristics like colors or materials. But I don't see why OpenCLIP features cannot encode these characteristics. And it's not well grounded to say that pre-trained OpenCLIP features are too complicated to model. As the author also mentioned the efficiency of discrete tokens, it's better to write the motivation clearly for using VQ-VAE.
- The writing could be improved:
    - the diffusion preliminary section is too abbreviated to serve as a separate section and is disjoined from Sec. 4.2.2 where the paper starts to relate to diffusion models. Then the semantic graph diffusion model employs a discrete diffusion model, that was proposed in previous work (Austin, et al., 2021). It seems that a large portion of Sec. 4.2.2 is not novel and could be treated as a preliminary study as well.
    - The annotation is overcomplicated and hard to follow. There are new definitions of previous variables throughout the method section, like $c, f$ in the first paragraph of Sec. 4.2, $\mathcal{C}_i$ and $\mathcal{F}_i$ in the first two lines of Sec. 4.2.2. $f$VQ-VAE is not a standard notation. There are also variables with omitted super/sub-scripts that are not explained clearly.

Some of the technical concerns might be entangled with writing issues of the current version. I am open to adjusting the rating if my concerns are resolved later.

**Questions:**

- What is the CLIP-aligned feature in Fig. 2? How is this feature vector applied to the VQ-VAE encoder?

- Though the relations are view-dependent, the generated instructions do not describe the viewpoint of the camera. Is there an assumed viewpoint for all scenes?

- Minor: in Fig. 5 row 4, the proposed methods generate a single bed while the instruction mentions a double bed.

Missing reference: LayoutGPT: Compositional Visual Planning and Generation with Large Language Models. Neurips 2023

---

> ### Author Response · Authors · 2023-11-15
> **Response to Reviewer qrjz - Part 1**
>
> Thank you for your valuable feedback. We are happy to know that you regarded the curated dataset as useful, and that the proposed method is effective and versatile. Below are our clarifications for your concerns. We value your input and would love to hear any follow-up you have to the response.
>
> **1. The novelty of the proposed method.**
>
> While acknowledging the proposed method relies on existing techniques, such as the diffusion model first introduced in 2015 [1] and vector quantization first proposed in 1984 [2], the novelty and contribution of our work stem from three key aspects:
>
> (1) we propose a new formulation of an instruction-driven scene synthesis pipeline, which integrates a semantic graph prior and a layout encoder, and notably improves the generation controllability; (2) our disentanglement design of semantic graph prior and layout decoder enables various useful applications in a zero-shot manner; (3) we curate a high-quality dataset to promote the research of language-driven 3D scene synthesis, which could be considered as a timely contribution that aligns with the current trends.
>
> Meanwhile, the basic idea behind the proposed method is to introduce structural priors for complex language-driven content generation. The proposed semantic graph prior is a general concept and can be directly applied to other tasks, such as 2D poster design. We hope the exploration of 3D scenes could inspire more research on other domains.
>
> **2. Some claims about using VQ-VAE require further substantiation.**
>
> OpenCLIP features indeed encode characteristics like colors and materials. However, these attributes are entangled within a single vector of 1280 dimensions. This high dimensionality makes it challenging to model these features together with other low-dimensional scene attributes, such as transitions and scales, which only have 3 dimensions each.
>
> The use of VQ-VAE is not for efficiency. Instead, it serves to simplify the learning task. Indexing discrete variables from a codebook can significantly reduce the burden of network optimization. Our preliminary experiments also suggest that both ATISS and DiffuScene face difficulties in modeling high-dimensional semantic feature distributions. To ensure fair comparisons in our experiments, we enhance these models to generate quantized features.
>
> **3.1. The diffusion preliminary section is too abbreviated and disjoined.**
>
> Making the preliminary diffusion models an independent section is because we apply two kinds of diffusion models in this work, i.e., discrete and continuous ones. Sec. 3 provides a general introduction to the core idea of diffusion models, which is agnostic of diffusion kernels, ranging from Gaussian to uniform categorical transitions and various other choices. By establishing this foundation in Sec. 3, we create a context for introducing the proposed semantic graph prior and layout decoder in a unified expression of diffusion models.
>
> It's important to note that both conditional diffusion models in this work, one for instruction-driven graph generation and the other for graph-conditional layout generation, are specifically tailored for respective tasks. They differ from the original ones proposed for images or texts, and are not suitable to be treated merely as a preliminary study.
>
> **3.2. The annotation is overcomplicated and hard to follow.**
>
> We apologize for any confusion caused by the annotation. We try our best to keep the annotations rigorous and consistent in the manuscript. We use lowercase characters like $c$ and $f$ for attributes of one object, and use calligraphic fronts like $\mathcal{C}$ and $\mathcal{F}$ for attributes of one scene. $\mathcal{C}_i$ and $\mathcal{F}_i$ refers to the specific $i$-th scene. We use superscripts to denote different scenes and subscripts to denote different objects in the same scene. Variables with omitted super-/sub-scripts mean a non-specific sample.
>
> We will try our best to keep the annotation concise in the revision of the manuscript. Thank you for bringing that to our attention.
>
> **4.1. What is the CLIP-aligned feature in Fig. 2?**
>
> The “CLIP-aligned feature” in Figure 2 refers to the output vector of OpenShape [3], which is a pretrained RGB point cloud encoder that is contrastively trained to be aligned with the output space of OpenCLIP ViT-bigG/14.
>
> **4.2. How is the CLIP-aligned feature vector applied to the VQ-VAE encoder?**
>
> As described in the first paragraph of Sec. 4.4, the encoder of $f$VQ-VAE is a stack of cross-attention modules. $n_f$ learnable tokens are employed as queries, and the CLIP-aligned feature vector is served as the key and value.
>
>
> ---
> [1] Deep Unsupervised Learning using Nonequilibrium Thermodynamics. ICML 2015.
>
> [2] Vector Quantization. IEEE Assp Magazine 1984.
>
> [3] OpenShape: Scaling Up 3D Shape Representation Towards Open-World Understanding. NeurIPS 2023.

---

> ### Author Response · Authors · 2023-11-15
> **Response to Reviewer qrjz - Part 2**
>
> **5. Is there an assumed viewpoint from all scenes?**
>
> The term "view-dependent" means that spatial relations are respected by the users who provide instructions. These spatial relations will change in accordance with the users' viewpoints. There is no need to specify the camera's perspective.
>
> **6. The proposed method generates a single bed while the instruction mentions a double bed.**
>
> Thank you for the careful assessment. Although we have significantly promoted the controllability of instruction-driven 3D scene synthesis, the iRecall scores of our method are still not 100%. Therefore, it is possible to observe incorrect triplets in the generated scenes. We hope follow-up works could further improve the performance.
>
> **7. Missing reference of LayoutGPT.**
>
> Thank you for the kind reminder. We will add this reference in the revision of this work.

---

> > ### Comment · Reviewer_qrjz · 2023-11-22
> > **Thank authors for the response**
> >
> > Thank you for your detailed response. I believe that the authors have well addressed my concerns. I have also read the paper several more times and other reviews as well. I am familiar with the tasks and datasets yet not an expert in the field. Therefore, I appreciate the patience of the authors in trying to address my concerns and questions. On the other hand, given the low dimensionality of the several predicted variables, I do feel that the task itself can be modeled in a more efficient and elegant way, especially when the final scene is composed by retrieving objects from an existing inventory. Despite saying so, I still value the contribution of this work and have raised my rating accordingly.

---

### Official Review · Reviewer_jsgt · 2023-11-02

**Soundness:** 3 good
**Presentation:** 3 good
**Contribution:** 3 good
**Rating:** 8
**Confidence:** 4

**Summary:**

The paper proposes a new scene generation framework with two stages. The first stage models the semantic graph, which consists of object categories, relations and appearances. With feature quantization, all these attributes become discrete variables, which can be learned through a discrete diffusion model. Conditioned on the semantic graph prior, the second stage uses a continuous diffusion model to learn the remaining scene attributes (rotations, translations and sizes). By modeling the semantic graph separately, the method enables various zero-shot downstream tasks. Experiments show that the proposed method achieves better scene generation results. The paper also creates a new dataset of text-scene pair for instruction-driven 3D scene synthesis.

**Strengths:**

- One main challenge of scene generation is the discrete and continuous nature of scene representation. The paper provides a new scene generation pipeline by disentangling the generation of discrete attributes and continuous attributes. Though the architecture designs of both the discrete and continuous diffusion model largely follow existing methods, the new formulation of the scene generation pipeline is novel. In addition, the disentanglement enables various zero-shot downstream tasks, improving the controllability of the generative model.
- The newly proposed dataset of scene-instruction pairs is a timely contribution that aligns well with the ongoing research of scene generation.
- The paper is clear and well-written, both detailed quantitative and qualitative results are provided.

**Weaknesses:**

Generally I think the paper is good. There are some questions in the part Questions.

**Questions:**

- More details about the dataset creation can be provided. It would be better if the author can provide some examples of the training data.
- In the semantic graph, relation e_{jk} can be determined by e_{kj}. Are the both directions of the instructions needed to be created? If so, how to determine which direction to use during training?
- What is the runtime of the proposed method compared to baselines?
- The instructions are generated mainly from rules. Do you think these rules can cover the majority of the instructions compared to a human-annotated dataset? What are the limitations of the instructions in the current dataset?

---

> ### Author Response · Authors · 2023-11-15
> **Response to Reviewer jsgt**
>
> Thank you for your insightful and valuable feedback. It is really inspiring to know that you think our formulation of scene generation is novel, effective and versatile, the curated dataset is a timely contribution, and the paper is clear and well-written. Below are our clarifications for your concerns. We value your input and would love to hear any follow-up you have to the response.
>
> **1. More details about the dataset creation can be provided.**
>
> The training data of instructions are very similar to those provided in the first column of Appendix Figure 5~7, which are for scenes in the test split, instead of the training split. The details about the data creation are provided in Appendix A. We also provide some examples below.
>
> (0) Object categories are provided by the original dataset: e.g., `{id1_category: "dressing_table"}`. (1) Spatial Relation Extraction: e.g., `(id1, “closely left”, id2)`. (2) Object Caption: e.g., `{id1_caption: "a 3D model of a wooden desk with a mirror on top in a white background"}`. (3) Caption Refinement: e.g., `{id1_refined_caption: "a modern wooden dressing table with a mirror on top"}`. (4) Instruction Generation: e.g., `{“Place”, “Add”, “Put”, …}` +  ` id1_refined_caption` + `{“closely to the left of”, “to the close left of”, “to the near left of”, …}` + `id2_refined_caption.` + `{Then, Next, And, ...}` + ...
>
> Our dataset and scripts for creation will be publicly available after the review period.
>
> **2. Are the both directions of the instructions needed to be created? If so, how to determine which direction to use during training?**
>
> We randomly pick one direction to generate instructions during training, e.g., “a chair is left to a table” and “a table is right of a chair”, where both instructions should correspond to the same relation in a scene. To incorporate this inductive bias, we did some tricks in the code. In particular, the final logits for loss computation are derived by averaging the upper triangular part and the symmetrized lower triangular part of the relation matrix.
>
> Our code will be publicly available after the review period.
>
> **3. What is the runtime of the proposed method compared to baselines?**
>
> In the default settings (T=100+10), our method takes about 12 seconds to generate a batch of 128 living rooms by our method on a single A40 GPU. In comparison, ATISS (Paschalidou et al., 2021) takes 3 seconds, and DiffuScene (Tang et al., 2023) requires 22 seconds.
>
> It's noteworthy that our method can be significantly accelerated by reducing the number of diffusion time steps. For instance, setting T=20+5 reduces the runtime to 3 seconds without a noticeable decline in performance. The impact of diffusion time steps is investigated in Sec. 5.5.1. We believe with more advanced diffusion techniques and in more complex scenes, diffusion models can be more effective and efficient than autoregressive models.
>
> **4. Can these rules cover the majority of the instructions compared to a human-annotated dataset? What are the limitations of the instructions in the current dataset?**
>
> While the curated instructions in our proposed dataset are derived from predefined rules, we believe that our model exhibits generalizability to a broader range of instructions. For example, in the stylization task, we utilize instructions in various sentence patterns such as "Let the room be wooden style" and "Make objects in the room black", as illustrated in Appendix Figure 8. We also experiment with instructions containing vague location words, like "Put a chair next to a double bed," wherein our method generates corresponding objects in all possible spatial relations (e.g., "left," "right," "front," and "behind").
>
> Nevertheless, our model still faces limitations in comprehending complex text instructions. For instance, (1) handling instructions with more required triplets, like 4 or 5, poses a challenge. (2) Additionally, identifying the same object within one instruction, such as "Put a table left to a sofa. Then add a chair to the table mentioned before" is also a difficult task. (3) Furthermore, InstructScene struggles with abstract concepts such as artistic style, occupants, and functionalities that do not occur in the curated instructions.
>
> These limitations are attributed to the CLIP text encoder, which is contrastively trained with image features and tends to capture global semantic information in sentences. Given the rapid development of large language models (LLMs), we believe the integration of LLMs into the proposed pipeline is a promising research topic.
>
> A viable approach to improve the quality of current instructions involves employing LLMs to refine entire sentences in the proposed dataset or using crowdsourcing to make the dataset curation pipeline semi-supervised. We hope the proposed dataset and creation pipeline could serve as a good starting point for creating high-quality instruction datasets.
>
> We will add this discussion to the revision of this work.

---

> > ### Comment · Reviewer_jsgt · 2023-11-23
> > **Response**
> >
> > Thank you for your response. My concerns are addressed. I recommend acceptance of this paper.

---

### Official Review · Reviewer_Peja · 2023-11-08

**Soundness:** 4 excellent
**Presentation:** 4 excellent
**Contribution:** 3 good
**Rating:** 8
**Confidence:** 5

**Summary:**

This paper proposes a framework that generates 3D indoor scenes from natural language prompts. The framework takes two stages. The first stage generates a scene graph from the language prompt. The scene graph contains category and vector quantized feature representation of objects and relations (as directed edges) between them. The second stage generates the scene layout from the scene graph, predicting the location, size and orientation of the objects. Both stage are learned with transformer-based diffusion models. Various techniques are adopted to allow using diffusion to learn discrete attributes and graphs.
Evaluation show that the proposed method outperforms previous state of the art methods (on slightly different tasks) convincingly. Qualitative examples appear to be of good quality as well.

**Strengths:**

- Very solid design choices. The authors clearly have good knowledge of techniques relevant to this problem, and the proposed solution make a lot of sense to me e.g. discretizing object features with VQVAE, discrete diffusion models, the use of MASK for diffusion-based graph generation, various learning tricks, etc.
- Good effort in creating a dataset suitable for this task (as existing scene datasets lack language descriptions). The pipeline for creating such text descriptions synthetically is reasonable.
- Clearly outperforms prior SOTAs over many metrics commonly used for this task.
- Qualitative results look good --- matches my impression of gt 3DFRONT scenes. Samples provided in the supp are realistic - exhibits many common minor issues, showing that they are clearly not cherry picked results (which is important considering the questionable reproducibility of many prior works).

**Weaknesses:**

- More of a system paper: combines many ideas from previous works on 3D scene generation and works on diffusion-based generation model. Seems that there is not too much truly new ideas here (and clarifications on where these happens would be great). I'm fine with it, to be honest, because this paper ticks my checkboxes for what a good system paper should be: clever/novel combination of ideas/components, solid execution, exceptional results. However, this is technically still a "weakness" I guess, as it limits the further applicability of the proposed method in other domains.
- Questionable choice of baselines. I get that previous graph-based/text-based works are not trained on 3D-FRONT, but it should be possible to compare with them in some capacities, or at least discussing the issues with them in the evaluation section,
- Despite the good discussions of issues of the way prior works handle latent object features (my impression was that they don't help much), I am left unconvinced that the proposed method actually make use of such features e.g. 1. would manually changing semantic feature f of an object result in notable difference in the final scene? 2. Can the model pick up stylistic information from the training data? Do objects that frequently occur together in the training data behave similarly in the generated scenes? 3. Is there any noticeable degrade in scene layout quality if semantic features are not encoded (i.e. just sample CAD models by category and size, which is a common practice for prior works) or if they are encoded with a smaller feature space?
- A user study would have been useful as it seems that humans, although not always consistent at rating the quality of the scenes, would be very good at checking if the generated scenes matches the text description.
- More discussions on how the qualitatively the proposed method works better than prior works would be very helpful. I can clearly that this is the case from the supplemental figures, and it would be great to highlight this in the main paper as well with a paragraph or so, as the proposed method clearly avoids many issues with prior works, especially on larger rooms. The set of quantitative metrics, while comprehensive, are known to be not the most sensitive metrics when it comes to evaluating indoor scenes, especially when it comes to layout/appearance details. More in depth analysis of the qualitative examples would make it a much stronger case that the proposed method is clearly superior.
- Lack of examples on how the proposed method can generalize & generate a diverse set of rooms from a single prompt. I think there can be many valid graphs for a single prompt and many valid layouts given a fixed semantic graph. It would be great to show examples that the proposed method can indeed achieve this.
- Most of the instructions are very specific, which makes sense consider how they are generated. However, in real applications, fuzzier instructions are also very important e.g. a room in the style of {certain style}, a room for {certain type of occupants}, a room that supports {certain functionalities}. Discussion on how to incorporating these instructions would be good.

**Questions:**

I think this is a good paper: it is clearly very solid technically, allows useful applications, and clearly outperforms prior works. I don't think there is anything that I need to see in the author responses. However, as state in the weakness section above, I do have issues here and there with the generalizability with the method, and with how the evaluations are carried out. If the authors could provide more insight regarding these concerns, then I would be much more inclined to champion this paper (which I currently am hesitant to do).

---

> ### Author Response · Authors · 2023-11-15
> **Response to Reviewer Peja - Part 1**
>
> Thank you for your insightful and valuable comments and suggestions. It is really inspiring to know that you think our work is solid for both design choices and creating a dataset, and good quantitative and qualitative performance. Below are our clarifications for your concerns. We value your input and would love to hear any follow-up you have to the response.
>
> **1. Applicability of the proposed method in other domains.**
>
> The basic idea behind the proposed method is to introduce structural priors for complex language-driven content generation. The proposed semantic graph prior is a general concept and can be directly applied to other tasks, such as 2D poster design. We hope the exploration of 3D scenes could inspire more research on other domains.
>
> **2. Discuss other previous graph-/text-based works that are not trained on 3D-FRONT.**
>
> Thank you for the kind reminder. There are plenty of great works on both graph-based and text-based 3D scene generation. We try our best to include them in the related work section (Sec. 2) and discuss the drawbacks of existing works.
>
> Two methods, namely ATISS (an autoregressive model) by Paschalidou et al. (2021) and DiffuScene (a diffusion model) by Tang et al. (2023), are selected as baselines in our work. That is because both of them are recently proposed (many papers in 2023 are still using the same architecture as ATISS [1, 2]), representative (autoregressive and diffusion models currently stand as two of the most popular generative models), and showcase state-of-the-art performance in various metrics. Other remarkable works, such as FastSynth [3], SceneFormer [4], DepthGAN [5] and Sync2Gen [6], despite their significance, demonstrated notably inferior performance compared to ATISS and DiffuScene, and most of them are unconditional generative methods, so they were excluded from our evaluation.
>
> We acknowledge that we may overlook some important works and would appreciate any suggestions. We will add them to the revision of this work.
>
> **3.1. Would manually changing semantic feature of an object result in notable differences in the final scene? Can the model pick up stylistic information from the training data? Do objects that frequently occur together in the training data behave similarly in the generated scenes?**
>
> We conduct two additional experiments to further validate our method: (1) masking the semantic feature of one object and utilizing a pretrained semantic graph prior for recovery; (2) masking semantic features of all objects except one and again using the pretrained semantic graph prior for recovery. Instructions for both experiments are set to none. Visualization results are presented in the supplementary material submitted during the rebuttal period.
>
> These results indicate the diversity of our method and highlight that semantic graph prior could effectively capture stylistic information and object co-occurrences from the training data. Our method trends to generate style consistent and thematic harmonious scenes, e.g., chairs and nightstands in a suit, and matched color palettes and cohesive artistic style. Other visualizations in the appendix also support this claim, especially with Figure 8 showcasing the generation of all semantic features based on both instructions and other scene attributes.
>
> **3.2. Is there any noticeable degradation in scene layout quality if semantic features are not encoded or if they are encoded with a smaller feature space?**
>
> We observed a significant decline in the appearance controllability and style consistency of generated scenes when semantic features were omitted. We include these degraded visualization results in the supplementary material.
>
> It arises from the fact that, without semantic features, the generative models solely focus on modeling the distributions of layout attributes, i.e., categories, translations, rotations, and scales. This exclusion of semantic features results in generated objects whose occurrences and combinations lack awareness of object style and appearance, which are crucial elements in scene design.
>
> We will add this discussion to the revision of this work.
>
>
> ---
> [1] MIME: Human-Aware 3D Scene Generation. CVPR 2023.
>
> [2] Learning 3D Scene Priors with 2D Supervision. CVPR 2023.
>
> [3] Fast and Flexible Indoor Scene Synthesis via Deep Convolutional Generative Models. CVPR 2019.
>
> [4] SceneFormer: Indoor Scene Generation with Transformers. 3DV 2021.
>
> [5] Indoor Scene Generation from a Collection of Semantic Segmented Depth Images. ICCV 2021.
>
> [6] Scene Synthesis via Uncertainty-driven Attribute Synchronization. ICCV 2021.

---

> ### Author Response · Authors · 2023-11-15
> **Response to Reviewer Peja - Part 2**
>
> **4. A user study would have been useful.**
>
> Thank you for this suggestion. However, we found applying a user study for evaluation could be tricky and potentially biased. Instead, we introduce two new quantitative metrics: (1) iRecall, measuring the proportion of the required triplets occurring in synthesized scenes to all provided in instructions and (2) $\Delta\coloneqq \frac{1}{N}\sum_{i=1}^{N}\text{CosSim}(\mathbf{f}_i,\mathbf{d}^{\text{style}}_i)-\text{CosSim}(\mathbf{f}_i,\mathbf{d}^{\text{class}}_i)$, designed to quantitatively assess if the generated scenes matches the required styles.
>
> Acknowledging the subjective nature of fidelity and aesthetics, we have presented comprehensive qualitative results in the appendix and leave it to readers to determine their preferences.
>
> **5. Discussions on how qualitatively the proposed method works better than prior works.**
>
> Thank you for this suggestion. In Appendix C.2, we discuss how our proposed method qualitatively outperforms prior works. We find that the autoregressive model ATISS tends to generate redundant objects, resulting in chaotic synthesized scenes. DiffuScene encounters challenges in accurately modeling object semantic features, often yielding objects that lack coherence in terms of style or pairing, thereby diminishing the aesthetic appeal of the synthesized scenes. Moreover, both of these baseline models frequently struggle to follow the provided instructions during conditional generation. In contrast, our approach demonstrates a notable capability to generate highly realistic 3D scenes that concurrently adhere to the provided instructions.
>
> We will add this discussion to the main paper in the revision of this work.
>
> **6. Examples on how the proposed method can generalize & generate a diverse set of rooms from a single prompt.**
>
> Thank you for this suggestion. We provide examples of a diverse set of scenes generated from a single prompt and the same semantic graph in the supplementary material, showcasing the diversity of the proposed generative method.
>
> We will add these visualization results in the revision of this work.
>
> **7. Discussion on how to incorporate fuzzier instructions.**
>
> While the curated instructions in our proposed dataset are derived from predefined rules, we believe that our model exhibits generalizability to a broader range of instructions. For example, in the stylization task, we utilize instructions in various sentence patterns such as "Let the room be wooden style" and "Make objects in the room black", as illustrated in Appendix Figure 8. We also experiment with instructions containing vague location words, like "Put a chair next to a double bed," wherein our method generates corresponding objects in all possible spatial relations (e.g., "left," "right," "front," and "behind").
>
> Nevertheless, our model faces challenges in comprehending abstract concepts such as artistic style, occupants, and functionalities that do not occur in the curated instructions. This limitation is attributed to the CLIP text encoder, which is contrastively trained with image features and tends to capture global semantic information in sentences. Given the rapid development of large language models (LLMs), we believe the integration of LLMs into the proposed pipeline is a promising research topic.
>
> A viable approach to improve the quality of current instructions involves employing LLMs to refine entire sentences in the proposed dataset or using crowdsourcing to make the dataset curation pipeline semi-supervised. We hope the proposed dataset and creation pipeline could serve as a good starting point for creating high-quality instruction datasets.
>
> We will add this discussion to the revision of this work.

---

> > ### Comment · Reviewer_Peja · 2023-11-22
> >
> > Thanks for the detailed response to my concerns.
> >
> > 1: More discussions on that, even some quick demonstrations, could be helpful here.
> >
> > 2: I'm more talking about works like Text2Scene, PlanIT, Scenegraphnet, etc. They are older, probably doesn't perform as well on the current dataset, but are still more straightforward comparisons. Even if you don't plan to compare against them (which is fine, I think their codes are not the easiest to adapt to 3D-FRONT), a few sentences of discussions on that could be helpful.
> >
> > 3.1 & 3.2: Thanks for the additional experiments. They are very convincing. Inclusion of a few of them in the main paper/supp would be great.
> >
> > 4. I don't quite buy the argument about user study being biased - I think they can be useful even with potential biases. However, I won't insist on having them - it's indeed a lot of work to design a proper user study and I think the current paper is strong enough without it anyways. The new metrics look good.
> >
> > 5. The discussions are great.
> >
> > 6. Fig 5 looks great. I am slightly concerned with Figure 6, seems that there are limited variations in layout, the semantic graph (I suppose) should define a larger space of possible layouts than presented in the figure? Might be helpful to check against the nearest neighbors in the training set, and also discuss this in the future work section if scene diversity conditioned on a single scene graph is indeed an issue.
> >
> > 7. Discussion makes sense. It is indeed challenging to generate the right type of synthetic data for these kind of instructions. The potential directions pointed out here seem to be good ones.
> >
> > Thanks again for the response - I will certainly argue for this paper's acceptance given this rebuttal.

---

### Author Response · Authors · 2023-11-15
**General Response**

We would like to express our gratitude to all the reviewers for their valuable, constructive, and thoughtful feedback. It is really inspiring to know that the majority of the reviewers consider that (1) the proposed method is meaningful, effective, solid and well-designed, (2) the quantitative and qualitative experiments are extensive, (3) the curated dataset is timely and useful, and (4) the paper is well-written and easy to follow.

We respond to all reviewers thoroughly in individual replies. We have also included more qualitative results in the supplementary material PDF to address concerns raised by some reviewers. We sincerely hope that our responses will be helpful and informative. If you still have concerns that we have not addressed, we would love to hear from you and please let us know how we can improve our work.

Thank you very much for your time and input!

---

### Meta-Review · Area_Chair_ce2k · 2023-12-06

**Metareview:**

**Summary**
- The paper proposes a method to generate 3D scenes via a learned semantic scene-graph prior and a layout decoder.  A discrete graph-based diffusion model is used to learn a distribution of semantic scene-graph conditioned on text-based user instructions.  The layout decoder is a diffusion model that generates the placement, size, and orientation of objects given a semantic scene-graph.  Experiments show the proposed model generates scenes that conforms better to the instructions compared to prior work modified to be conditioned on text input.  To allow for the text conditioning, a dataset of paired scenes and instructions is also contributed.

**Strengths**
- All reviewers are positive on this work, find the method to be "well-designed" (rQD7) with "solid design choices" (Peja)
- The paper also contribute a dataset of paired scenes and instructions.  This is currently lacking in the community.
- Experiments are "extensive" (rQD7) and show that the proposed method outperforms prior work, the qualitative results are also compelling and representative (does not look to be overly cherry-picked Peja).
- The paper is also well-written and easy to follow (jsgt, rQD7)

**Weaknesses**
- The main weakness of the work is the lack of an user study (Peja)

**Suggested improvements to the paper**

In the author response and discussion with reviewers, the authors indicated that they will revise the manuscript with additional information to address reviewer concerns.  Some of the information and visualizations requested by reviewers are provided as part of the supplemental materials, but not yet integrated into the main paper or appendix.  The AC recommends the authors integrate the supplement into the appendix and make sure that the main paper references the appendix appropriately.

Other points that should be added in the paper include:
- Discussion of handling of arbitrary instructions (rQD7, Peja)
- Discussion of limitations of the instructions in the current dataset (jsgt)
- Some examples of training data (jsgt)
- Runtime of the proposed method (jsgt)
- Add discussion of relevant related work pointed out by the reviewers (qrjz)
  LayoutGPT: Compositional Visual Planning and Generation with Large Language Models [Feng et al. 2023]
- Improve discussion of other previous graph-/text-based works that are not trained on 3D-FRONT (Peja) - The difference to prior work with scene-graphs can be discussed a bit more. It should be discussed which of the prior text to scene work also uses graph representation (e.g. Chang et al. Ma et al.)
- Make sure that notation is clear and easy to follow (qrjz)
- Fix minor typos:
  Table 4: "Bebind" => "Behind"

**Justification For Why Not Higher Score:**

- While the approach may be applicable to other areas beyond scene generation, it is not that clear.  The scene generation community is a relatively narrow sub-community within ICLR and the work may not have impact beyond the scene generation community.

**Justification For Why Not Lower Score:**

- Reviewers are all in favor of accepting the work.  There is increasing interest in the community in text-to-3D scene generation, and paper is well-executed with the proposed method is sufficiently novel to be highlighted.

---

### Decision · Program_Chairs · 2024-01-16

Accept (spotlight)